# Comprehensive analysis of the human ESCRT-III-MIT domain interactome reveals new cofactors for cytokinetic abscission

**Dawn M Wenzel**[1]*[†‡], **Douglas R Mackay**[2†], **Jack J Skalicky**[1†], **Elliott L Paine**[1], **Matthew S Miller**[1], **Katharine S Ullman**[2]*, **Wesley I Sundquist**[1]*

[1]Department of Biochemistry, University of Utah School of Medicine, Salt Lake City, United States; [2]Department of Oncological Sciences, Huntsman Cancer Institute, University of Utah, Salt Lake City, United States

**Abstract** The 12 related human ESCRT-III proteins form filaments that constrict membranes and mediate fission, including during cytokinetic abscission. The C-terminal tails of polymerized ESCRT-III subunits also bind proteins that contain Microtubule-Interacting and Trafficking (MIT) domains. MIT domains can interact with ESCRT-III tails in many different ways to create a complex binding code that is used to recruit essential cofactors to sites of ESCRT activity. Here, we have comprehensively and quantitatively mapped the interactions between all known ESCRT-III tails and 19 recombinant human MIT domains. We measured 228 pairwise interactions, quantified 60 positive interactions, and discovered 18 previously unreported interactions. We also report the crystal structure of the SPASTIN MIT domain in complex with the IST1 C-terminal tail. Three MIT enzymes were studied in detail and shown to: (1) localize to cytokinetic midbody membrane bridges through interactions with their specific ESCRT-III binding partners (SPASTIN-IST1, KATNA1-CHMP3, and CAPN7-IST1), (2) function in abscission (SPASTIN, KATNA1, and CAPN7), and (3) function in the 'NoCut' abscission checkpoint (SPASTIN and CAPN7). Our studies define the human MIT-ESCRT-III interactome, identify new factors and activities required for cytokinetic abscission and its regulation, and provide a platform for analyzing ESCRT-III and MIT cofactor interactions in all ESCRT-mediated processes.

## Editor's evaluation

The importance of the information provided to the community, their quality and the new questions they open will certainly make this paper an essential step forward in understanding the regulation of ESCRT-III.

**\*For correspondence:**
dwenzel@mcw.edu (DMW);
katharine.ullman@hci.utah.edu (KSU);
wes@biochem.utah.edu (WIS)

[†]These authors contributed equally to this work

**Present address:** [‡]Department of Biochemistry, Medical College of Wisconsin, Milwaukee, United States

**Competing interest:** The authors declare that no competing interests exist.

## Introduction

To complete cell division, a cell must equally and fully partition a faithfully duplicated genome into two nascent progeny cells through mitosis, and these cells must physically separate through cytokinesis. In animal cells, cytokinesis initiates when an actomyosin contractile ring ingresses to create a cleavage furrow between the two spindle poles following chromosome segregation (*Addi et al., 2018*). The plasma membrane furrow ultimately narrows into an intercellular membrane bridge containing a dense central array of microtubules (the midbody), which then undergoes abscission to create two new cells. Premature abscission in the presence of aberrantly segregated DNA can lead to tension-induced double strand breaks in chromatin bridges that traverse the cleavage furrow (*Janssen et al., 2011*), chromosome instability (*Umbreit et al., 2020*), and formation of micronuclei and extensive DNA damage through chromothripsis (*Zhang et al., 2015*; *Crasta et al., 2012*). Hence, abscission is

required for cell proliferation and its misregulation can contribute to DNA damage and even cancer development (*Pharoah et al., 2013*; *Sadler et al., 2018*).

The ESCRT (Endosomal Sorting Complexes Required for Transport) pathway mediates abscission in cultured mammalian cells (*Carlton and Martin-Serrano, 2007*; *Morita et al., 2007*), and in neural progenitor cells in vivo (*Tedeschi et al., 2020*; *Little et al., 2021*). Approximately 30 human ESCRT factors comprise five subcomplexes: ALIX, ESCRT-I, ESCRT-II, ESCRT-III, and VPS4, which assemble sequentially to constrict and sever membranes (*McCullough et al., 2018*). During cytokinesis, ESCRT complexes are recruited to the central Flemming body within the midbody by the CEP55 adaptor protein. CEP55 binds the early-acting ESCRT protein, ALIX (*Lee et al., 2008*; *Christ et al., 2016*; *Morita et al., 2007*; *Carlton and Martin-Serrano, 2007*), which in turn recruits ESCRT-III subunits of the CHMP4 family (*Christ et al., 2016*; *McCullough et al., 2008*; *Katoh et al., 2003*; *Katoh et al., 2004*; *Kim et al., 2005*). In a parallel pathway, CEP55 binds the TSG101 subunit of ESCRT-I/II complexes (*Morita et al., 2007*; *Carlton and Martin-Serrano, 2007*; *Christ et al., 2016*; *Lee et al., 2008*), again leading to ESCRT-III recruitment. ESCRT-III recruitment is also promoted by the midbody protein SEPT9 (*Karasmanis et al., 2019*), and other CEP55-independent pathways (*Merigliano et al., 2021*; *Addi et al., 2020*). Humans express 12 homologous ESCRT-III proteins that fall into eight different families, termed CHMP1-7 and IST1. The different ESCRT-III subunits co-polymerize to form spiraling filaments within the midbody (*Guizetti et al., 2011*; *Mierzwa et al., 2017*; *Pfitzner et al., 2021*; *Nguyen et al., 2020*; *Elia et al., 2011*). These filaments recruit VPS4 AAA+ ATPases that in turn promote dynamic ESCRT-III subunit exchange, midbody constriction, membrane fission, and daughter cell separation (*Pfitzner et al., 2021*; *Mierzwa et al., 2017*; *Elia et al., 2012*; *Pfitzner et al., 2020*).

Abscission timing and progression are regulated by the evolutionarily conserved abscission/NoCut cell cycle checkpoint (*Norden et al., 2006*; *Steigemann et al., 2009*). This checkpoint pauses abscission progression so that upstream mitotic events can be completed correctly before the cell irreversibly separates. Several different conditions are known to sustain abscission checkpoint activation, including incompletely segregated DNA within the midbody (*Norden et al., 2006*; *Mendoza and Barral, 2008*; *Amaral et al., 2016*), nuclear pore subunit depletion (*Mackay et al., 2010*), DNA damage resulting from replication stress (*Mackay and Ullman, 2015*), and midbody tension (*Lafaurie-Janvore et al., 2013*; *Strohacker et al., 2021*; *Caballe et al., 2015*). These conditions all converge on phosphorylated Aurora B (pAurB) kinase, which maintains checkpoint signaling and inhibits ESCRT activity. AurB is targeted to the intercellular bridge as a component of the multi-protein Chromosomal Passenger Complex (CPC), where it phosphorylates multiple substrates, including the regulatory ESCRT-III protein CHMP4C (pCHMP4C) (*Carlton et al., 2012*; *Capalbo et al., 2012*; *Steigemann et al., 2009*). pCHMP4C associates with ANCHR, and this complex sequesters VPS4 AAA+ ATPases from abscission sites, thereby inhibiting abscission progression (*Thoresen et al., 2014*). pAurB and pCHMP4C also localize to cytoplasmic Abscission Checkpoint Bodies (ACBs), where pro-abscission machinery, including ALIX and ESCRT-III proteins, are sequestered from the midbody (*Strohacker et al., 2021*). The importance of the abscission checkpoint and its regulation by CHMP4C is underscored by the discovery that a missense mutation in the penultimate CHMP4C residue (A232T) disrupts ALIX binding, inactivates the checkpoint, and predisposes carriers to several types of cancers (*Sadler et al., 2018*; *Pharoah et al., 2013*).

In addition to constricting the midbody, ESCRT-III filaments also recruit cofactors that contribute to abscission and checkpoint regulation. Human ESCRT-III proteins have conserved helical core domains that mediate filament formation (*McCullough et al., 2018*; *Pfitzner et al., 2021*; *Bajorek et al., 2009b*), and variable C-terminal tails that contain short peptide elements called MIT-Interacting Motifs (MIMs) (*Figure 1A*). MIMs bind cofactors that contain MIT (Microtubule-Interacting and Trafficking) domains (*Hurley and Yang, 2008*; *Figure 1B*). MIT domains are simple three helix bundles, but they can bind MIM elements in at least seven distinct ways (*Skalicky et al., 2012*; *Obita et al., 2007*; *Scott et al., 2005*; *Stuchell-Brereton et al., 2007*; *Kieffer et al., 2008*, *Yang et al., 2008*; *Solomons et al., 2011*; *Fujioka et al., 2014*), which we have termed Type 1–7 binding modes (*Figure 1C*).

Humans express more than 20 MIT domain-containing proteins, which also contain a variety of associated activities that could function in cytokinesis (*Figure 1B*). Well-characterized examples of MIT domain-containing proteins that bind ESCRT-III proteins and perform important midbody functions include: (1) VPS4. The archetypal MIT domains of the related VPS4A and VPS4B ATPases bind promiscuously to different ESCRT-III filaments. These interactions promote assembly of VPS4 hexamers and

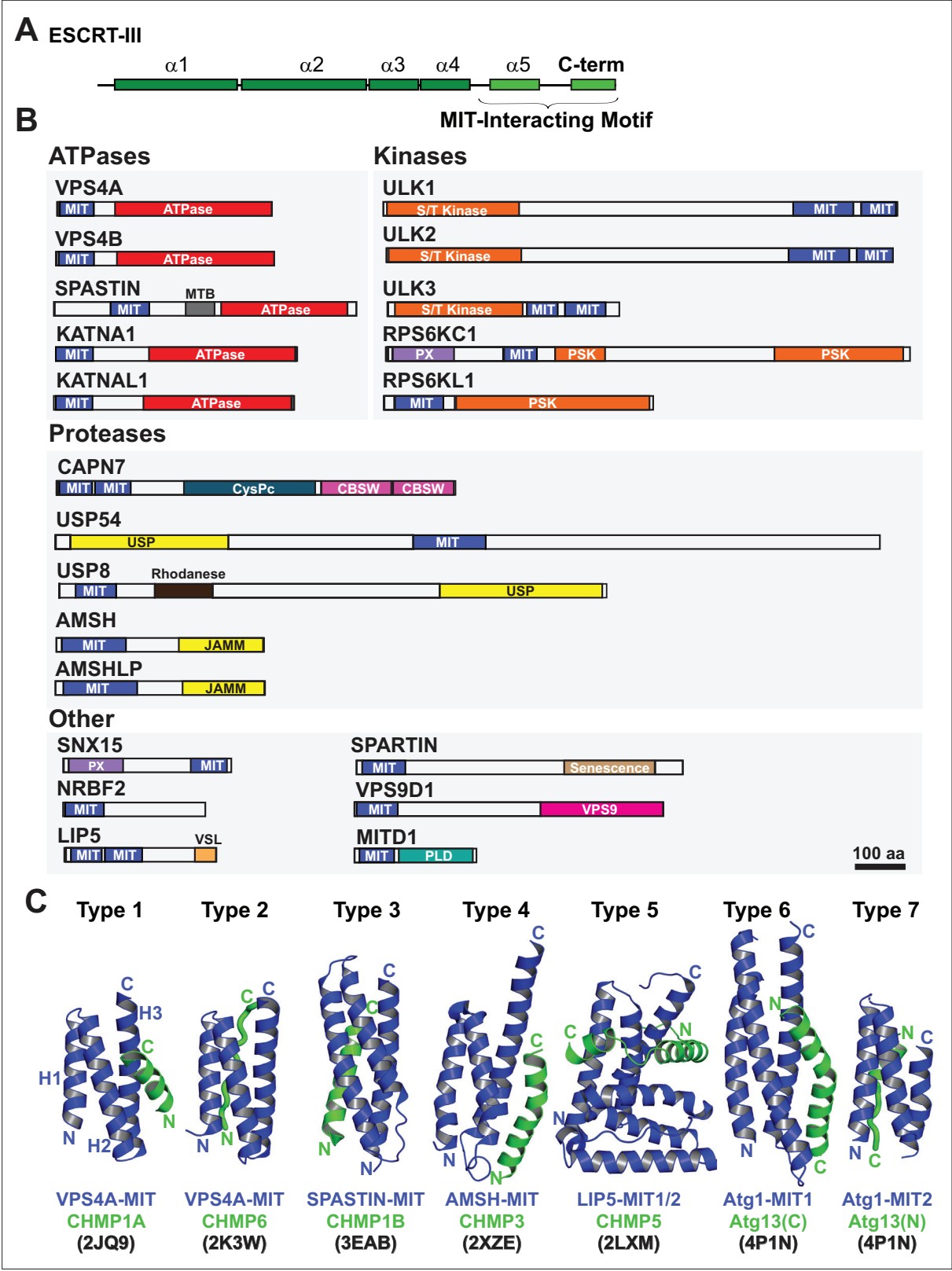

**Figure 1.** Domain organization of ESCRT-III and MIT domain-containing proteins. (**A**) Generalized ESCRT-III schematic, depicting the conserved helical core domain and the variable C-terminal tail that contains MIT-interacting Motif (MIM) elements. (**B**) Human proteins with MIT domains, grouped by enzymatic categories. Three dimensional structures (PDB IDs in parenthesis) are available for the MIT domains from human VPS4A (1YXR, 2JQ9, 2K3W), VPS4B (2JQH, 2JQK, 4U7Y, 1WRO, 2CPT), SPASTIN (3EAB), USP8 (2A9U), AMSH (2XZE), NRBF2 (4ZEY), LIP5 (2LXL, 2LXM, 4TXP, 4TXQ, 4TXR, 4U7E),

*Figure 1 continued on next page*

*Figure 1 continued*

SPARTIN (4U7I), MITD1 (4A5X), and ULK3 MIT2 (4WZX). Abbreviations: MIT, Microtubule Interacting and Trafficking; MTB, Microtubule Binding Domain; S/T Kinase, Serine/Threonine Kinase domain; PSK, Pseudokinase domain; PX, Phosphoinositide binding domain; CysPc, Calpain protease domain; USP, Ubiquitin Specific Protease Domain; PLD, Phospholipase D-like domain; VSL, Vta1-SBP1-LIP5 domain; JAMM, JAB1/MPN/Mov34 metalloenzyme domain; CBSW, calpain-type beta-sandwich domain. (**C**) Gallery showing the different types of binding interactions formed between MIT domains (blue) and MIM elements (green). For clarity, the three helix bundles of the different MIT domains are shown in approximately equivalent orientations. *Type 1 interaction:* a helical MIM binds in the groove between MIT helices 2 and 3 (H2/H3 groove), and is oriented parallel to MIT helix 3. *Type 2 interaction:* the MIM forms an extended strand that binds in the MIT H1/H3 groove, parallel to MIT helix 3. *Type 3 interaction:* a helical MIM binds in the MIT H1/3 groove, parallel to MIT helix 3. *Type 4 interaction:* Similar to Type 1, except that the longer MIM helix binds lower in the H2/H3 groove and interacts with the H2/H3 loop. *Type 5 interaction:* Two MIM helices and adjacent linkers wrap nearly completely around the MIT domain. *Type 6 interaction:* a mixed helix/strand MIM element binds in the MIT H1/3 groove, antiparallel to MIT helix 3. *Type 7 interaction:* a helical MIM element binds in the MIT 2/3 groove, antiparallel to helix 3. The specific complexes shown in the figure are labeled below, together with their PDB accession codes.

activate ATPase activity (*Han et al., 2015*; *Azmi et al., 2008*; *Norgan et al., 2013*). VPS4 MIT domains can alternatively bind a MIM element within ANCHR, which sequesters the ATPases from the abscission zone (*Thoresen et al., 2014*). (2) ULK3 kinase. The second ULK3 MIT domain binds IST1, which localizes this kinase to the midbody where it phosphorylates ESCRT-III proteins and inhibits membrane fission (*Caballe et al., 2015*). (3) SPASTIN. The MIT domain of the microtubule severing AAA+ ATPase SPASTIN binds CHMP1B (*Yang et al., 2008*), which reportedly localizes the enzyme to the midbody where it clears spindle microtubules to allow abscission (*Connell et al., 2009*). (4) MITD1. The MIT domain of MITD1 binds preferentially to CHMP1A, CHMP1B, CHMP2A and IST1 and localizes this phospholipase-D family member to midbodies where it stabilizes the bridge, preventing abscission failure (*Hadders et al., 2012*; *Lee et al., 2012*). (5) SPARTIN. The MIT domain of the hereditary spastic paraplegia protein SPARTIN preferentially binds IST1, which localizes the protein to the midbody to support abscission (*Renvoisé et al., 2010*).

The 12 ESCRT-III and >20 MIT human proteins have the potential to form a complex combinatorial network of ESCRT-III-MIT interactions that help mediate and regulate abscission. However, the ESCRT binding and cytokinetic functions of many human MIT proteins have not yet been tested, and prediction of MIT-MIM interactions can be difficult owing to their remarkable variety of different possible binding modes (*Figure 1C*). To address these limitations, we systematically quantified the binding of all 12 human ESCRT-III MIM elements to nearly all known human MIT domains. Three MIT enzymes with interesting activities and ESCRT-III binding patterns were further tested for midbody localization, and roles in cytokinetic abscission and the abscission checkpoint. These studies revealed that SPASTIN, KATANIN-P60 (KATNA1), and CALPAIN-7 (CAPN7) all localize to midbodies through specific MIT-ESCRT-III interactions and are required for efficient abscission, and that SPASTIN and CAPN7 are also required for abscission checkpoint maintenance. These studies define the human MIT-ESCRT-III interactome and identify new factors and activities required for cytokinetic abscission and its regulation.

## Results
### MIT-MIM interaction screen

We defined the ESCRT-III-MIT interactome by using fluorescence polarization anisotropy (FP) to quantify the binding interactions between recombinant human MIT domains and ESCRT-III MIM peptides. These experiments employed fluorescently labeled C-terminal peptides (52–88 residues) that encompass the known MIM elements of the 12 human ESCRT-III proteins (*Figure 1A*, *Supplementary file 1B*). The ESCRT-III tails were fluorescently labeled at their N-termini. Literature analyses were used to identify 21 candidate human MIT domains (*Ciccarelli et al., 2003*; *Rigden et al., 2009*; *Skalicky et al., 2012*; *Xiao et al., 2008*; *Fujioka et al., 2014*; *Figure 1B*; see Materials and methods), which were expressed as recombinant proteins and purified to homogeneity. Constructs encoding the tandem MIT domains of LIP5, CAPN7, and ULK3 spanned both domains, thereby allowing intramolecular interactions to occur and native structures to form. (*Skalicky et al., 2012*; *Vild et al., 2015*; *Guo and Xu, 2015*). 19/21 of the human MIT domain constructs could be purified in sufficient quantities for FP binding titrations (*Figure 2*). The two exceptions were the C-terminal tandem MIT domains of ULK1

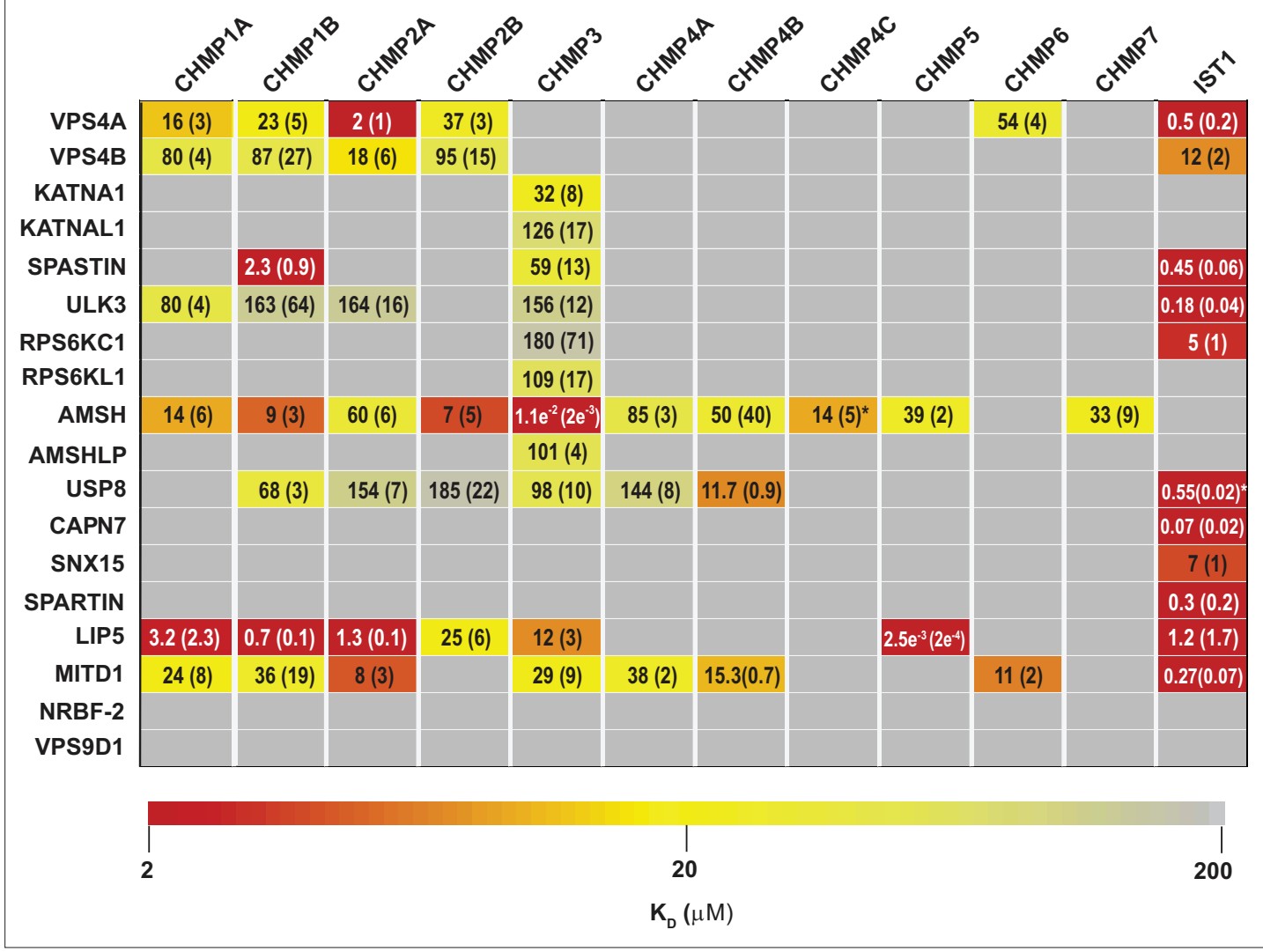

**Figure 2.** ESCRT-III-MIT domain interaction network. Equilibrium dissociation constants (μM) for pairwise binding interactions are displayed for each ESCRT-III-MIT pair and are reported as averages of at least three independent measurements. Values in parenthesis denote ± standard deviation. Interactions are color-coded based on strength of measured binding affinity (see key bar). Asterisks denote $K_I$ values that were determined from competition experiments (see Materials and methods and *Figure 2—figure supplement 5B-E*). Binding constants for ULK3(MIT)$_2$ and IST1 are from *Caballe et al., 2015*, but were measured in the same fashion as the other interactions and are reproduced here for comparison.

The online version of this article includes the following source data and figure supplement(s) for figure 2:

**Figure supplement 1.** Binding isotherms for MIT domains that bind promiscuously to ESCRT-III protein tails.

**Figure supplement 2.** Binding isotherms for MIT domains that bind specifically to ≤3 different ESCRT-III tails.

**Figure supplement 3.** USP54 MIT binds weakly to all ESCRT-III C-terminal tails.

**Figure supplement 4.** Raw binding data for MIT domains that do not bind any ESCRT-III tails.

**Figure supplement 5.** Competitive binding analyses of CHMP4C tails binding to MIT domains.

**Figure supplement 6.** ESCRT-III binding is not conserved across ULK family members.

**Figure supplement 6—source data 1.** Annotated uncropped western blots and raw images for *Figure 2—figure supplement 6A and B*.

and ULK2, which function in autophagy (*Fujioka et al., 2014*) and apparently do not bind ESCRT-III proteins (see below).

Pairwise FP binding isotherms were fit to 1:1 equilibrium binding models (*Figure 2*, *Figure 2—figure supplements 1–4*, *Figure 3*). Dissociation constants ($K_D$) for the 228 interactions tested are summarized in *Figure 2* and color-coded based upon interaction strengths. Our screen generally recapitulated binding interactions reported previously using orthogonal techniques (See Supplemental

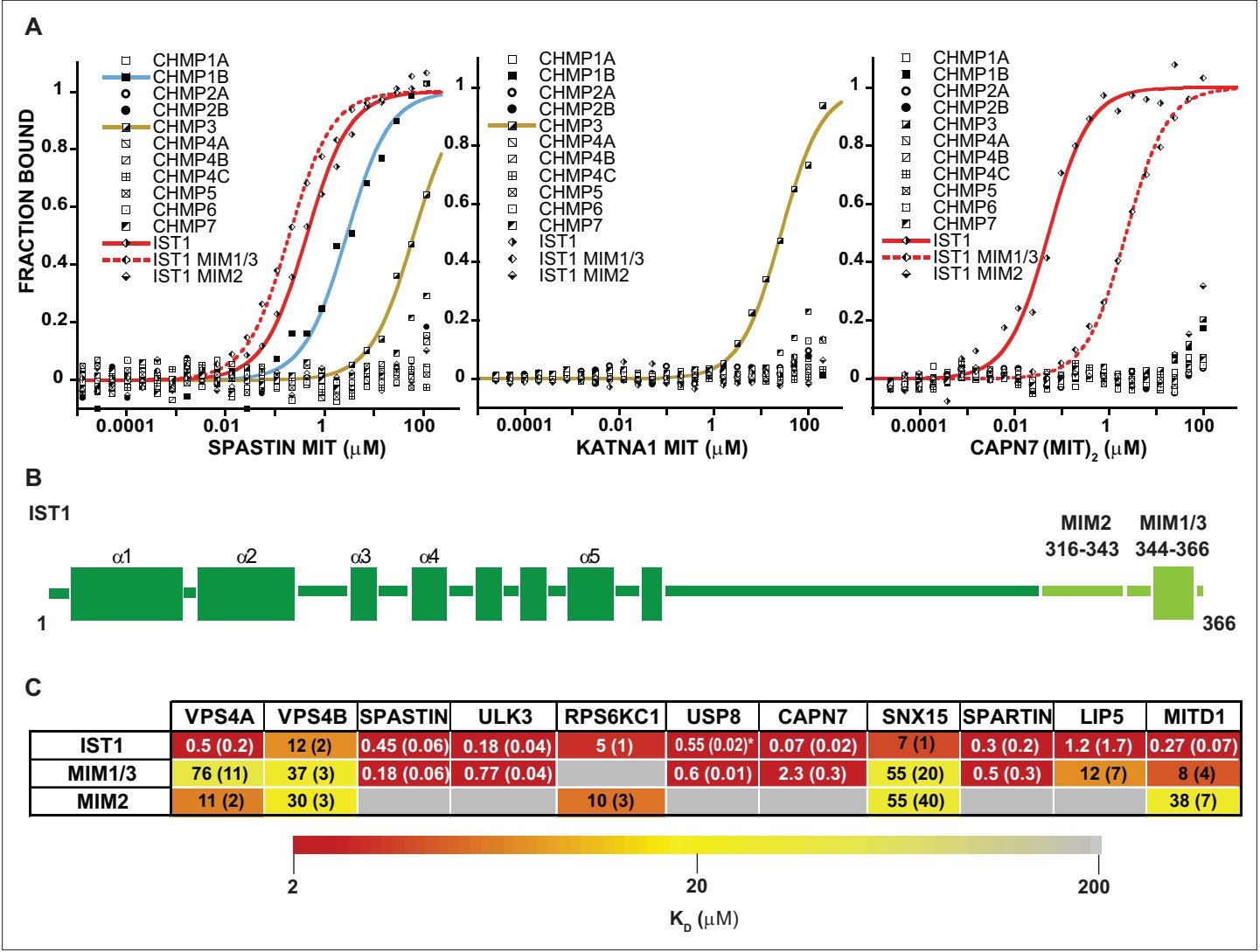

**Figure 3.** MIT-MIM Binding Interactions of SPASTIN, KATNA1 and CAPN7 and IST1. (**A**) Binding isotherms from human ESCRT-III C-terminal tails and the MIT domains from SPASTIN (left), KATNA1 (middle), and CAPN7 (right). Representative binding data are shown for each ESCRT-III-MIT pairwise binding interaction. Binding isotherms with $K_D$ <200 µM are fit with colored curves. Weak and non-binders ($K_D$ >200 µM) are plotted as black and white points. (**B**) Domain schematic of the ESCRT-III protein IST1 showing the position of the Type 2, MIM$_{316-343}$ and Type 1/3, MIM$_{344-366}$ elements within the full-length protein. Data for the 'IST1' peptide encompasses residues 316–366. (**C**) Dissociation constants for MIM$_{344-366}$ and MIM$_{316-343}$ peptides with IST1-binding MIT domains. Dissociation constants shown are averages calculated from at least three independent experiments with standard deviation reported in parentheses. Interactions are color-coded based on strength of measured binding affinity (see key bar). Asterisk denotes $K_I$ values determined from competition experiments (see Materials and methods and *Figure 2—figure supplement 5E*). Binding data for ULK3(MIT)$_2$ and IST1, IST1-MIM$_{344-366}$ and IST1-MIM$_{316-366}$ are from *Caballe et al., 2015*, but are reproduced here for comparison.

Discussion), but we cannot rule out the possibility that in some cases the position of the label could have interfered with or artificially enhanced binding interactions for some ESCRT-III-MIT pairs. Pairwise binding isotherms for weak binding pairs often did not reach saturation (*Figure 2—figure supplements 1–3*), and the $K_D$s for those interactions should therefore be considered approximations. In cases where isotherms did not reach half saturation at the highest MIT concentration tested (usually 100 µM), we did not attempt to estimate the $K_D$.

Our binding survey confirmed a series of previously reported ESCRT-III-MIT domain interactions (See Supplemental discussion), and also revealed 18 previously unreported interactions (*Figure 2*), including establishing new direct links between ESCRT-III proteins and the MIT domains from KATNA1, KATNAL1, RPS6KC1, RPS6KL1, SNX15, and AMSHLP. These new interactions should facilitate future investigations of ESCRT-associated activities.

## ESCRT-III binding behaviors

MIT domains displayed a range of different ESCRT-III binding behaviors that we classified as promiscuous (>3 ESCRT-III binding partners) (*Figure 2—figure supplement 1*), specific (≤3 ESCRT-III binding partners) (*Figure 3A*, *Figure 2—figure supplement 2*), non-specific (*Figure 2—figure supplement 3*), and non-binding (*Figure 2—figure supplement 4*). The seven promiscuous binders bound ESCRT-III tails with high ($K_D$ <2 μM) or moderate ($K_D$ <100 μM) affinities. The MIT domain of the deubiquitinase AMSH exhibited the greatest promiscuity, binding all but two of the 12 different ESCRT-III subunits. The eight specific binders interacted with CHMP3 (KATNA1, KATNAL1, RPS6KL1, RPS6KC1, and AMSHLP), IST1 (RPS6KC1, CAPN7, SNX15, and SPARTIN), or CHMP1B, CHMP3, and IST1 (SPASTIN). The predicted MIT domain of the catalytically inactive deubiquitinase, USP54 (*Rigden et al., 2009*), bound weakly ($K_D$ >100 μM) to nearly all ESCRT-III proteins, indicating that these interactions were probably non-specific (*Figure 2—figure supplement 3*). Finally, two MIT domains (NRBF2 and VPS9D1) did not bind any ESCRT-III proteins (*Figure 2—figure supplement 4*). The MIT domain of NRBF2 has been characterized structurally (PDB 4ZEY, 2CRB) and shown to interact with the VPS15 kinase to promote autophagosome biogenesis (*Young et al., 2019*). The putative VPS9D1 MIT domain was predicted through bioinformatics, has not been characterized structurally, and has no known ESCRT connections (*Rigden et al., 2009*).

The tightest MIM-MIT binding pair was LIP5-CHMP5 ($K_D$ = 2.5 nM), which reflects the fact that CHMP5 tail helices 5 and 6 and adjacent linkers form an amphipathic 'leucine collar' that wraps almost completely around the second LIP5 MIT domain (*Figure 1C*; *Skalicky et al., 2012*).

The ESCRT-III proteins IST1, CHMP3, and CHMP1B partnered with the largest subset of MIT domains (*Figure 2*). IST1 bound 11/19 MIT domains with high affinity, and this promiscuity likely reflects the presence of two MIMs in the IST1 tail; a helical Type 1 or Type 3 (Type 1/3) element (residues 344–366) and an extended Type 2 element (residues 316–343) (*Bajorek et al., 2009a*, *Guo and Xu, 2015*; *Figure 3B*). To distinguish the contributions of each IST1 MIM element to MIT-binding, we tested their binding separately to each of the IST1-interacting MIT domains, using shorter peptides comprising either MIM (*Figure 3B and C*). These experiments revealed that the IST1 Type 1/3 (MIM$_{344-366}$) element bound 10/11 MIT domains, and the Type 2 element (MIM$_{316-343}$) bound 5/11 MIT domains. Three binding patterns for the IST1 MIMs emerged, with MIT domains binding: (1) only the Type 1/3 element (SPASTIN, ULK3, USP8, and SPARTIN), (2) only the Type 2 element (RPS6KC1), or (3) both MIMs (VPS4A, VPS4B, CAPN7, SNX15, LIP5, and MITD1) (*Figure 3C*). Although we did not directly detect binding between the IST1 Type 2 element and the LIP5 and CAPN7 MIT domains, the IST1 Type 2 element appeared to contribute to LIP5 and CAPN7 binding because its absence reduced binding affinity >10 fold (vs. the full IST1 tail). Thus, both IST1 MIM elements can bind MIT domains, thereby contributing to promiscuity. Moreover, both MIM elements can simultaneously engage single MIT proteins in some cases, thereby increasing binding affinity (*Osako et al., 2010*; *Bajorek et al., 2009a*).

At the other end of the spectrum, a fluorescently labeled CHMP4C peptide did not bind detectably to any MIT domain tested. This was surprising, particularly as MITD1, USP8, and AMSH all bound the other two CHMP4 paralogs, CHMP4A and CHMP4B (*Figure 2*). CHMP4C is unique in containing a serine-rich insertion that is phosphorylated by AurB (*Carlton et al., 2012*; *Capalbo et al., 2012*; *Figure 2—figure supplement 5A*). We reasoned that this flexible insert might reduce the fluorescence anisotropy in bound complexes, thereby generating false negative results. To test this idea, we assayed whether the CHMP4C peptide could competitively inhibit AMSH, MITD1 and USP8 MIT binding to labeled CHMP4B (*Figure 2—figure supplement 5B*). These experiments revealed that CHMP4C bound AMSH MIT ($K_i$ 14 μM), but still failed to bind MITD1 or USP8. CHMP4C performs a unique role in abscission checkpoint regulation (*Carlton et al., 2012*; *Capalbo et al., 2012*), and this specialized function may have relieved the selective pressure to maintain some MIT binding interactions.

As noted above, we were unable to express and purify the ULK1 or ULK2 tandem MIT domains in sufficient quantities for direct binding assays. Nevertheless, we felt it was important to test whether the MIT domains from these related kinases could bind ESCRT-III proteins because: (1) the MIT domains of yeast and human ULK1/2 kinases bind ATG13 within autophagosome initiation complexes (*Chan et al., 2009*; *Fujioka et al., 2014*), and (2) the MIT domain of the related ULK3 kinase binds tightly to IST1, and weakly to CHMP1A, CHMP1B, CHMP2A, and CHMP3 (*Caballe et al., 2015*)

and (*Figure 2*). We therefore performed pulldown assays from human HEK239T cell extracts to test whether ULK1(MIT)$_2$ could bind ESCRT-III proteins (*Figure 2—figure supplement 6A and B*). Positive control pulldown assays recapitulated the known ULK3(MIT)$_2$-ESCRT-III interactions, as well as ULK1(MIT)$_2$-ATG13 binding. However, ULK1(MIT)$_2$ did not bind detectably to any of the ESCRT-III binding partners of ULK3, implying that these ESCRT-III interactions are not conserved in the ULK family of protein kinases.

In summary, most MIT proteins can engage ESCRT-III proteins and their interactions span a range of specificities and affinities, indicating that they couple to ESCRT pathway functions in a variety of different ways.

## SPASTIN, KATNA1, and CAPN7 as paradigms for MIT-ESCRT-III interactions

Three MIT proteins with specific ESCRT-III interactions and interesting associated enzymatic domains were selected for further characterization: (1) SPASTIN (IST1, CHMP1B, and CHMP3 binding), (2) KATNA1 (CHMP3 binding only), and (3) CAPN7 (IST1 binding only) (*Figure 3A*). ESCRT-III interactions with SPASTIN (*Reid et al., 2005*; *Yang et al., 2008*; *Agromayor et al., 2009*) and CAPN7 *Yorikawa et al., 2008*; *Osako et al., 2010* have been described previously, whereas SPASTIN and KATNA1 binding to CHMP3 have not. The single MIT domain of SPASTIN bound tightly to the IST1 MIM$_{344-366}$ (Type 1/3 binding) and CHMP1B (Type 3) C-termini. The CAPN7 tandem MIT domain binds both MIM elements (*Osako et al., 2010*), and in this case multiple different binding modes are possible. SPASTIN and KATNA1 are microtubule severing AAA+ ATPases (*Sharp and Ross, 2012*). Both have been implicated in clearing midbody microtubules prior to abscission (*Yang et al., 2008*; *Connell et al., 2009*; *Benz et al., 2012*; *Matsuo et al., 2013*), but neither has been linked to abscission checkpoint activity. CAPN7 is an understudied cysteine protease that has not previously been linked to any cytokinetic functions.

## Structure and analysis of the SPASTIN MIT-IST1 complex

The SPASTIN MIT domain is unusual in binding specifically to three different ESCRT-III proteins: CHMP1B, CHMP3, and IST1. The SPASTIN MIT-CHMP1B complex has been characterized structurally (*Yang et al., 2008*; *Figure 1C*), whereas the SPASTIN MIT-IST1 and SPASTIN-CHMP3 interactions have not. We determined a high resolution (1.15 Å) crystal structure of the SPASTIN MIT-IST1 complex using a SPASTIN MIT$_{112-196}$ construct. (*Figure 4A*, and *Figure 4—figure supplement 1A and B*, and *Supplementary file 3*; PDB 7S7J). The structure revealed that the IST1 MIM$_{344-366}$ adopts an amphipathic helix that buries its hydrophobic side chains in the SPASTIN MIT H1/H3 groove (Type 3 binding, *Figure 4A*, *Figure 4—figure supplement 1*). The IST1 and CHMP1B MIM elements exhibit similar binding modes (*Figure 4A–D*), but with two significant differences (*Figure 4—figure supplement 1C*): (1) The longer CHMP1B helix extends three additional turns beyond the IST1 N-terminus, and (2) the MIT H1/H3 groove expands to accommodate the longer CHMP1B helix, with a maximal displacement of 2.5 Å at the N-terminus of helix 3. The structure of SPASTIN-IST1 is nearly identical to SPARTIN-IST1, which also adopts a Type 3 interaction (*Guo and Xu, 2015*; *Figure 4—figure supplement 1D*). IST1 binding is slightly tighter than CHMP1B binding, likely owing to enhanced hydrophobic interactions with the SPASTIN H1/H3 groove (*Figure 4—figure supplement 1C*). IST1 buries two aromatic and two aliphatic side chains, whereas the CHMP1B binding element lacks aromatic residues and buries only three aliphatics.

Based on our structural analyses, we created SPASTIN MIT mutations designed to disrupt the binding of both CHMP1B and IST1 (F124D, red in *Figure 4A–D*) and to disrupt only CHMP1B binding without affecting IST1 binding (L177D, cyan in *Figure 4A–D*). As shown in *Figure 4E and F*, these mutations behaved as designed, thereby providing a set of mutants that we could use to compare the biological effects of disrupting SPASTIN binding to both CHMP1B and IST1 vs. specifically disrupting binding to CHMP1B alone. CHMP3 binding was also inhibited by both SPASTIN F124D and L177D mutants. Thus, CHMP3 also likely binds as an extended Type 3 helix in the SPASTIN MIT H1/H3 groove (*Figure 4—figure supplement 2*), and only IST1 retained binding to the SPASTIN L177D mutant owing to its shorter Type 3 helix.

## KATNA1 and CAPN7 binding to ESCRT-III proteins

We also screened for MIT point mutations that could block ESCRT-III binding to KATNA1 and CAPN7 by making a series of disruptive mutations in the MIT H1/H3 and H2/H3 grooves and testing whether

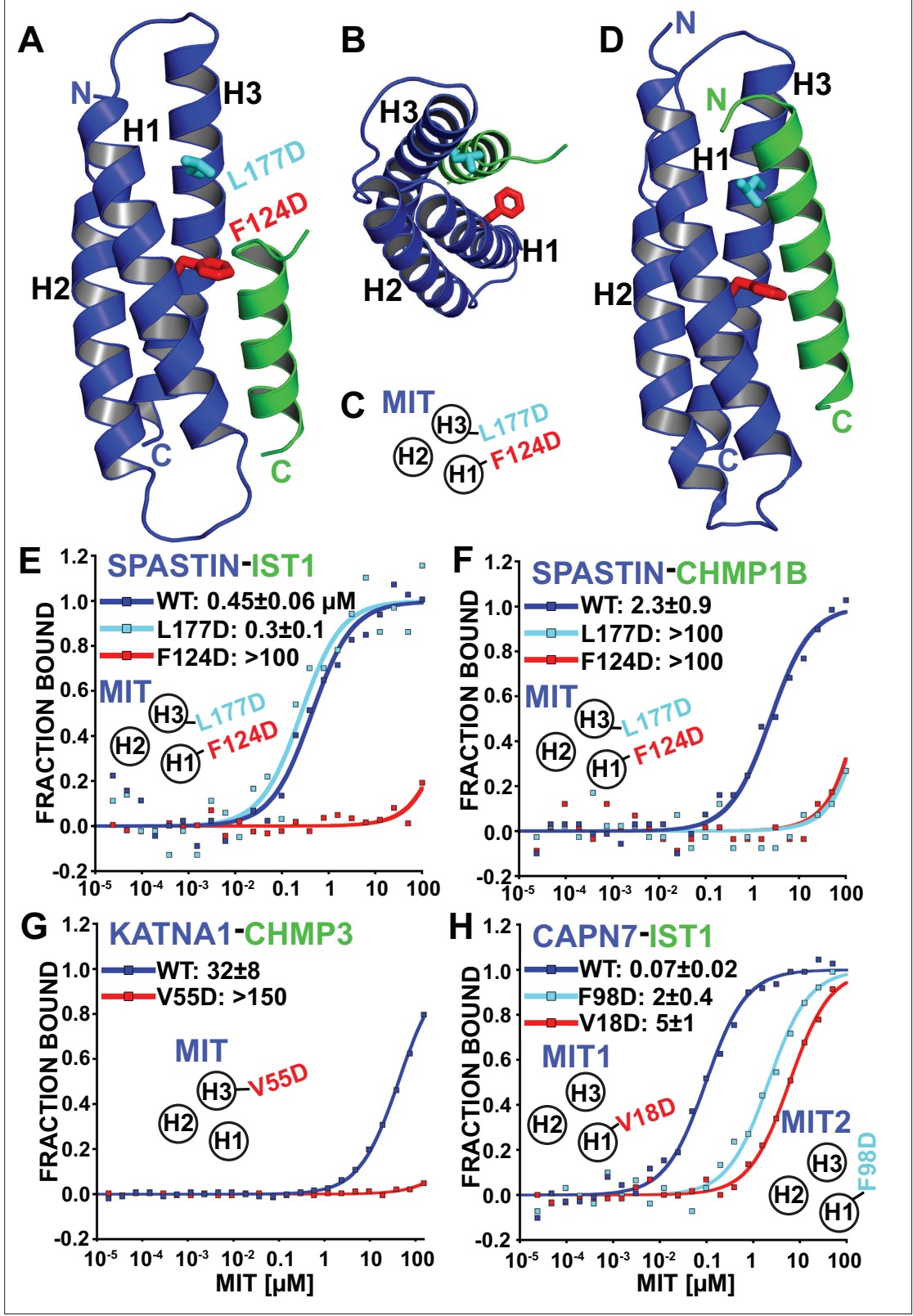

**Figure 4.** Identification of MIT-MIM binding mutants. (**A and B**) Two orientations of the SPASTIN MIT-IST1 Type 3 structure (PDB 7S7J) displaying the locations of binding mutants (stick representation). See *Figure 4—figure supplement 1A and B*, for a detailed view of the interface and *Supplementary file 3* for data collection and structure refinement statistics. (**C**) Cartoon depicting the positions of mutation sites in the SPASTIN MIT H1/H3 groove. (**D**) SPASTIN-CHMP1B structure (PDB 3EAB) showing the location of binding mutants (stick representation). See , *Figure 4—figure*

*Figure 4 continued on next page*

*Figure 4 continued*

*supplement 1C* for an overlay of both SPASTIN structures. (**E–G**) Binding isotherms for WT and mutant MIT domains with the designated MIM elements. (**E**) SPASTIN-IST1. (**F**) SPASTIN-CHMP1B. (**G**) KATNA1-CHMP3, and (**H**) CAPN7-IST1. For E and H, binding was measured using IST1 peptides composed of both MIM elements (residues 316–366). MIT groove binding cartoons within the figures show locations of mutations in the relevant MIT domain. See also *Figure 4—figure supplement 2* for binding isotherms for WT and mutant SPASTIN MIT domains with the CHMP3 MIM element.

The online version of this article includes the following figure supplement(s) for figure 4:

**Figure supplement 1.** Analysis of SPASTIN-IST1 complex and comparisons with SPASTIN-CHMP1B and SPARTIN-IST1.

**Figure supplement 2.** Binding isotherms for WT and mutant SPASTIN MIT domains with the CHMP3 MIM element.

these mutations abrogated binding in our fluorescence polarization anisotropy binding assay. In the KATNA1-CHMP3 case, a point mutation in the MIT H1/H3 groove (V55D) eliminated CHMP3 binding (*Figure 4G*). Our data therefore again indicate a Type 3 interaction for the helical CHMP3 MIM element. CAPN7 has tandem MIT domains, and we found that IST1 binding was significantly reduced (~30–70 fold) by point mutations in the H1/H3 groove of either the first (V18D) or second (F98D) CAPN7 MIT domains (*Figure 4H*). Together with our MIM mapping experiments (*Figure 3C*), these data indicate that both CAPN7 MIT domains engage both IST1 MIM elements, employing Type 2 (MIM$_{316-343}$) and Type 3 (MIM$_{344-366}$) binding modes.

## SPASTIN, KATNA1, and CAPN7 localize to midbodies

As an initial screen for cytokinetic functions, we examined whether endogenous SPASTIN, KATNA1, and/or CAPN7 localize to the midbodies of dividing cells (*Figure 5A–C*, respectively). Midbodies were co-stained for microtubules and CEP55, which is recruited to either side of the Flemming body where it initiates ESCRT complex assembly (*Fabbro et al., 2005*; *Carlton and Martin-Serrano, 2007*; *Morita et al., 2007*). Antibody staining specificity was confirmed in parallel experiments in which each MIT protein was separately depleted using siRNA (*Figure 5A–C*; right panels).

In midbody-containing cells, all three MIT proteins co-localized with CEP55 ring structures on either side of the Flemming body (*Figure 5A–C*). Uniquely, KATNA1 also distributed along the midbody arms, reminiscent of localization observed for caveolae, and termed 'midbody entry points' (*Andrade et al., 2022*). Arm staining was particularly prominent in early midbody-stage cells (*Figure 5B*), and a similar KATNA1 staining pattern has been reported in rat cell lines (*Matsuo et al., 2013*). Our observations of SPASTIN and KATNA1 Flemming body localization are also in good agreement with previous reports (*Yang et al., 2008*; *Connell et al., 2009*; *Matsuo et al., 2013*). Importantly ours is the first report that CAPN7 localizes to midbodies.

MIT protein localization was also examined in cells in which abscission checkpoint signaling was sustained by nucleoporin depletion (*Strohacker et al., 2021*; *Figure 5D–F*). In these experiments, cells were synchronized using thymidine treatment and release, together with siRNA depletion of NUP153 and NUP50 (siNups), which sustains the abscission checkpoint. In all cases, the MIT proteins again localized to Flemming bodies, although in each case abscission checkpoint signaling led to more diffuse staining and spreading of the signals to midbody arms. Thus SPASTIN, KATNA1, and CAPN7 localize to midbodies under both unperturbed and checkpoint active conditions, where they are poised to function in abscission and/or the abscission checkpoint.

## SPASTIN, KATNA1, and CAPN7 function in abscission

To test for abscission functions, we depleted SPASTIN, KATNA1, and CAPN7 and quantified abscission failure, as reflected by increased numbers of multi-nucleated cells and cells with persistent intercellular bridges (*Figure 5G* and *Figure 5—figure supplement 1*). Specific depletion and abscission phenotypes were confirmed using two different siRNA oligonucleotides in each case, and successful target protein depletion was confirmed by Western blot (*Figure 5—figure supplement 1*).

Cells lacking KATNA1 and CAPN7 both showed strong abscission failure phenotypes, with near doubling of the percentages of cells with midbodies or multiple nuclei (vs. control cells transfected with non-targeting (NT) siRNA). These pronounced phenotypes resembled the abscission defects observed with moderate knockdown of the essential IST1 protein (positive control) (*Bajorek et al., 2009a*, *Agromayor et al., 2009*). Cells lacking SPASTIN also exhibited significant abscission defects, although the effects were weaker. Our results are consistent with previously reported roles for KATNA1

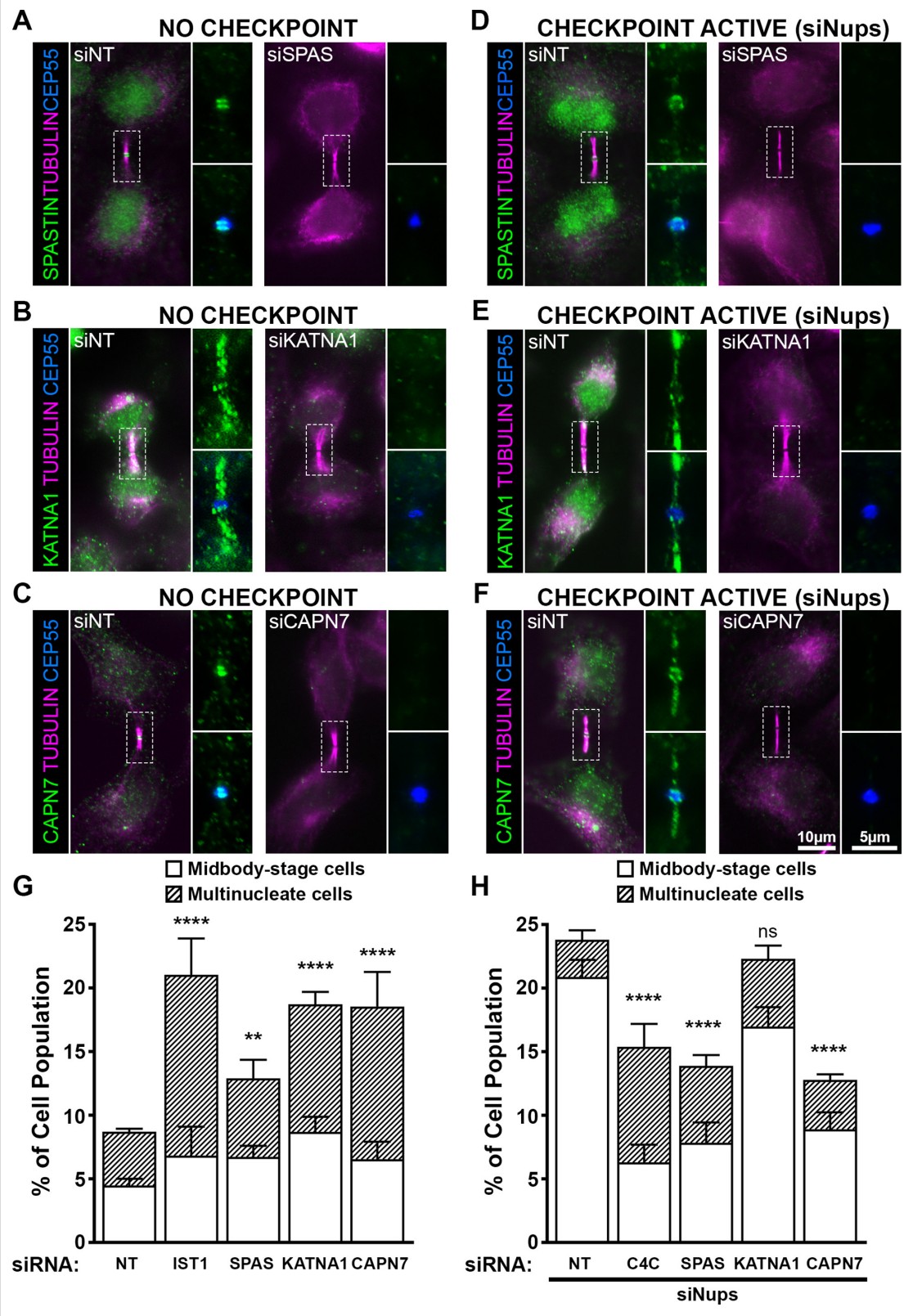

**Figure 5.** ESCRT-III-interacting proteins SPASTIN, KATNA1, and CAPN7 are recruited to the midbody and are differentially required for abscission and maintenance of the abscission checkpoint. Immunofluorescence images of SPASTIN (**A, D**), KATNA1 (**B, E**), and CAPN7 (**C, F**) after treatment with the indicated siRNAs. Checkpoint Active samples (**D–F**) were generated by additional treatment with siNups combined with a thymidine synchronization/release. Antibodies specific for α-TUBULIN and CEP55 were used to identify midbody-stage cells and the Flemming body, respectively. Enlargements of

*Figure 5 continued on next page*

*Figure 5 continued*

selected regions are shown at the right of each image, with the target protein shown alone (**top**) or together with CEP55 (**bottom**). (**G, H**) Quantification of abscission defects (midbody-stage cells and multinucleate cells) under asynchronous conditions (**G**) or with a sustained abscission checkpoint (**H**). Bars represent the average and standard deviation from n=5 independent experiments where N>500 cells were counted per experiment. Statistical analysis was performed using ANOVA, comparing total abscission defects (Midbody-stage cells + Multinucleate cells) after each individual siRNA treatment to siNT. ****p<0.0001, ***p<0.0005, **p<0.02, ns = not significant.

The online version of this article includes the following source data and figure supplement(s) for figure 5:

**Figure supplement 1.** Confirmation of the efficiency and specificity of protein depletion by siRNA treatments.

**Figure supplement 1—source data 1.** Annotated uncropped western blots and raw images for *Figure 5—figure supplement 1A and B*.

and SPASTIN in promoting abscission by clearing midbody spindle microtubules from abscission sites (*Matsuo et al., 2013*; *Yang et al., 2008*; *Connell et al., 2009*). The observation that CAPN7 is required for efficient cytokinetic abscission is a new discovery.

## SPASTIN and CAPN7 are required for abscission checkpoint maintenance

We also tested whether SPASTIN, KATNA1, or CAPN7 were required to maintain the AurB-mediated abscission checkpoint. As described above, co-depletion of the nuclear pore proteins NUP153 and NUP50 sustains abscission checkpoint activity, leading to an accumulation of midbody-connected cells (*Mackay et al., 2010*). However, simultaneous co-depletion of either SPASTIN or CAPN7 significantly reduced midbody accumulation, indicating roles for both proteins in abscission checkpoint maintenance (*Figure 5H*, *Figure 5—figure supplement 1*). Indeed, the effects of depleting either SPASTIN or CAPN7 were at least as penetrant as depleting CHMP4C (positive control), which plays a well-characterized role in inhibiting abscission in response to checkpoint signaling (*Capalbo et al., 2012*; *Carlton et al., 2012*). Unlike SPASTIN or CAPN7 depletion, KATNA1 depletion did not significantly alter midbody numbers, although a second oligo targeting KATNA1 showed a modest but significant decrease in cells undergoing an abscission checkpoint arrest (*Figure 5H*, *Figure 5—figure supplement 1*). Thus, our data indicate that SPASTIN and CAPN7 are required to sustain abscission checkpoint arrest, and KATNA1 may also contribute weakly.

## ESCRT-III proteins recruit SPASTIN, KATNA1, and CAPN7 to midbodies

To unite our binding and functional data, we tested whether ESCRT-III interactions were responsible for recruiting SPASTIN, KATNA1 and CAPN7 to function at midbodies. This was done by generating cell lines that expressed doxycycline (DOX)-inducible, siRNA-resistant mCherry constructs fused to wildtype (WT) and mutant MIT proteins. We then treated with siRNA to deplete the respective endogenous MIT proteins (*Figure 6—figure supplement 1*) while inducing expression of the mCherry fusion proteins and imaging the cells to test for midbody localization (*Figure 6*). As in *Figure 5*, cells were also synchronized with sustained checkpoints to maximize the number of midbody-stage cells (*Strohacker et al., 2021*). Importantly, all three WT mCherry fusion constructs recapitulated localization of the endogenous MIT counterpart within the midbody (*Figure 6*). The percentages of "arms only" localization patterns were higher for the endogenous constructs in every case, however, potentially reflecting reduced antibody epitope accessibility at the protein-rich Flemming body. We found some additional differences between expression constructs and their individual endogenous counterparts, such as a larger population of "Flemming Body + Arms" in mCherry-SPASTIN cells, which again may reflect epitope exposure (*Figure 6A*). These observations suggest that there may be more variation in SPASTIN localization than currently appreciated, particularly when the abscission checkpoint is sustained. For purposes here, however, this assay provides a robust system in which to probe the requirements for ESCRT-III-mediated recruitment to the midbody.

Like WT mCherry-SPASTIN, the L177D mutant SPASTIN localized to more than half of all midbodies (*Figure 6A and B*). The proportion of midbodies occupied by SPASTIN may reflect a temporally restricted role and/or be influenced by overall sensitivity of the assay. Strikingly, however, the F124D mutant did not localize to midbodies. These data imply that ESCRT-III binding is required to localize SPASTIN to the midbody. The SPASTIN F124D mutation disrupts binding to CHMP1B, IST1, and CHMP3 whereas the L177D mutation selectively permits only IST1 binding (*Figure 4*, *Figure 4—figure*

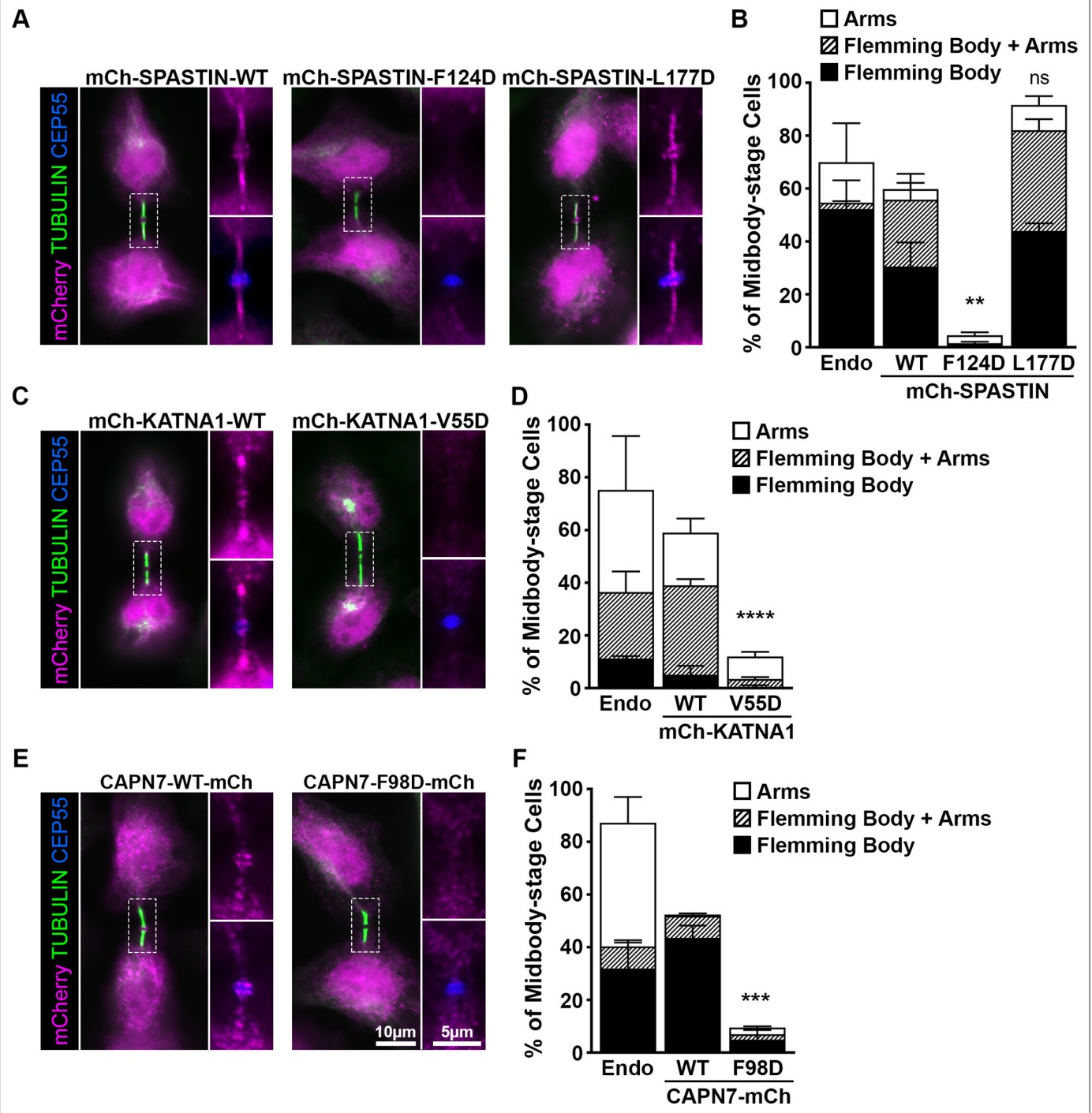

**Figure 6.** ESCRT-III binding to SPASTIN, KATNA1, and CAPN7 is required for midbody localization when the abscission checkpoint is sustained. (**A**) Immunofluorescence of DOX-inducible cell lines expressing siRNA-resistant mCherry-SPASTIN-WT, mCherry-SPASTIN-F124D, and mCherry-SPASTIN-L177D constructs under sustained abscission checkpoint conditions (see Materials and methods). Endogenous SPASTIN was depleted by siRNA treatment of the DOX-inducible cell lines and induced protein expression/localization was detected using an anti-mCherry antibody. Antibodies to α-TUBULIN and CEP55 were used to identify midbody-stage cells and the Flemming Body, respectively. Enlargements of selected regions are shown at the right of each image with the target protein shown alone (**top**) or together with CEP55 (bottom). (**B**) Quantification of midbody localization for endogenous SPASTIN from *Figure 5* (Endo) and the indicated DOX-inducible cell lines. Bars are the average and standard deviation from three independent experiments where >100 midbody-stage cells were counted. Statistical analysis was performed using ANOVA, comparing total midbody localization of the F124D and L177D mutants to WT. (**C, D**) Immunofluorescence and quantification of midbody localization for endogenous KATNA1 (*Figure 5*) and DOX-inducible cell lines expressing siRNA-resistant mCherry-KATNA1-WT and mCherry-KATNA1-V55D constructs as in A and B,

*Figure 6 continued on next page*

*Figure 6 continued*

except that endogenous KATNA1 was depleted by siRNA in the DOX- inducible cell lines. Statistical analysis was performed using an unpaired t-test, comparing the V55D mutant to WT. (**E, F**) Immunofluorescence and quantification of midbody localization for endogenous CAPN7 (*Figure 5*) and DOX-inducible cell lines expressing siRNA-resistant CAPN7-mCherry and CAPN7-F98D-mCherry constructs as in A and B, except that endogenous CAPN7 was depleted by siRNA in the DOX-inducible cell lines. Statistical analysis was performed using an unpaired t-test comparing the F98D mutant to WT. ****p<0.0001, ***p=0.0005, **p=0.0024, ns = not significant.

The online version of this article includes the following source data and figure supplement(s) for figure 6:

**Figure supplement 1.** Confirmation of the efficiency and specificity of protein depletion by siRNA and DOX-inducible protein expression.

**Figure supplement 1—source data 1.** Annotated uncropped western blots and raw images for *Figure 6—figure supplement 1* (**A–C**).

**Figure supplement 2.** The KATNA1 V55D mutation does not disrupt KATNB1 binding.

**Figure supplement 2—source data 1.** Annotated and uncropped western blots and raw images for *Figure 6—figure supplement 2B*.

*supplement 2*). Therefore, our data imply that CHMP1B and CHMP3 binding are dispensable for midbody localization when IST1 binding capability is maintained.

The KATNA1 MIT domain binds the C-terminal domain of the KATNB1 subunit to form the KATNA1 holoenzyme (*Faltova et al., 2019*; *Rezabkova et al., 2017*). KATNB1 activates and localizes KATNA1 to centrosomes where it severs mitotic spindle microtubules (*Matsuo et al., 2013*). However, KATNB1 does not appear to be responsible for localizing KATNA1 for cytokinesis function, as the two proteins reportedly do not co-localize within the midbody (*Matsuo et al., 2013*). It has therefore not been clear how KATNA1 is recruited to the midbody to function in abscission. Our localization data demonstrate that WT mCherry-KATNA1 localizes to Flemming bodies and midbody arms, but the V55D mutant does not (*Figure 6C and D*). CHMP3 is the sole ESCRT-III binding partner of KATNA1, and the V55D mutation disrupts this partnership (*Figure 3*), implying that CHMP3 localizes KATNA1 to the midbody for abscission.

The structure of the KATNA1 MIT-KATNB1 heterodimer reveals that V55 sits adjacent to the KATNB1 binding site (*Rezabkova et al., 2017*; *Figure 6—figure supplement 2A*), and we therefore tested the possibility that the V55D mutation might also disrupt the canonical KATNA1-KATNB1 partnership. This was not the case, however, as KATNB1 co-precipitated with full length KATNA1 WT and V55D mutant proteins equally well from HEK293T cell extracts (*Figure 6—figure supplement 2B*). In contrast, a KATNA1 MIT mutation known to disrupt KATNB1 binding, R14A (*Rezabkova et al., 2017*), abolished KATNB1 binding, demonstrating that the pulldown assay was sensitive to specific disruption. Hence, our data imply that the KATNA1 V55D mutation discriminates between CHMP3 and KATNB1 binding, and therefore that CHMP3 is likely responsible for localizing KATNA1 to Flemming bodies. Consistent with this model, CHMP3 localizes to Flemming bodies (*Mierzwa et al., 2017*; *Dukes et al., 2008*), and its depletion increases the propensity of cells to develop multiple nuclei and midbody bridges (*Morita et al., 2010*), thereby phenocopying KATNA1 loss (*Figure 5G*).

Finally, our MIT screening data suggested that IST1 could be responsible for localizing CAPN7 to midbodies (*Figure 2*). In support of this model, a point mutation in the second CAPN7 MIT domain that decreased IST1 binding ~30-fold (F98D, see *Figure 3*) also potently suppressed CAPN7 targeting to the Flemming body (*Figure 6E and F*). Thus, in all three cases we examined in detail, our MIT screening data identified potential ESCRT-III binding partners and point mutations that blocked binding, and in every case the point mutations also abolished MIT protein midbody localization.

## Discussion

We have comprehensively surveyed the MIT-ESCRT-III interactions between human ESCRT-III and MIT domain-containing proteins. We tested 228 pairwise interactions between MIT domains and ESCRT-III C-terminal tails, observed 60 positive interactions, and discovered 18 new interactions (*Figures 2 and 3*, see Supplemental discussion). We found that most human MIT domains can mediate ESCRT-III binding, further implicating MIT proteins as a major class of ESCRT pathway cofactors.

SPASTIN, KATNA1, and CAPN7 served as paradigms for using the binding data to discover new ESCRT cofactors that function in cytokinesis. Our screening assays identified ESCRT-III binding partners and interaction sites (*Figure 3*), enabled new structure determinations (*Figure 4*, *Figure 4—figure supplement 1*), and supported mutational analyses of MIT protein midbody localization (*Figure 4*,

*Figure 6*). Together with previous studies, our work indicates that the primary biological function of MIT domains is to localize proteins to sites of ESCRT-driven membrane remodeling. MIT domains have also evolved to mediate protein-protein interactions in other pathways, most notably in auto-phagosome initiation (*Fujioka et al., 2014*; *Young et al., 2019*). Collectively, MIT-MIM interactions constitute a complex recognition network, and several principles that emerged from our studies are highlighted below. A more comprehensive discussion of the different interactions is provided in the Supplemental discussion.

## MIT recruitment by ESCRT-III polymers

Although we studied the pairwise interactions of MIT domains and ESCRT-III tails, these interactions likely occur predominantly in the context of polymeric ESCRT-III filaments. Polymerization creates the potential for high avidity binding, and variations in ESCRT-III subunit compositions can alter binding modes and tune binding affinities. These properties are important because subunit compositions change as filaments mature and constrict membranes (*Pfitzner et al., 2021*; *Banjade et al., 2021*; *Pfitzner et al., 2020*). Current models for ESCRT-III assembly, derived from studies in simpler yeast systems, hold that CHMP6 initially nucleates the assembly of ESCRT-III strands composed of CHMP4 subunits. CHMP4 strands then recruit partner strands composed of CHMP2/CHMP3 subunits (*Teis et al., 2008*; *Babst et al., 2002*), which are subsequently exchanged for CHMP1/IST1 subunits (*Pfitzner et al., 2020*). Hence, the cadre of recruited MIT protein binding partners may evolve as filaments mature.

These different principles are nicely illustrated by VPS4 enzymes and their associated activator LIP5 (Vta1 in yeast), which form 'supercomplexes' that display 30 MIT domains; six from the VPS4 hexamer and 24 from the 12 associated $LIP5(MIT)_2$ elements (*Skalicky et al., 2012*; *Yang et al., 2012*). VPS4 MIT domains bind promiscuously to many ESCRT-III tails, and although binding is often weak avidity effects can presumably create high affinity binding. We measured strong to moderate, Type 1 VPS4 MIT binding to the late- acting CHMP1, CHMP2 and IST1 subunits (nM to mid-µM $K_D$s), with three- to 20-fold higher affinities consistently seen for VPS4A MIT vs. VPS4B MIT. CHMP3 binding was below our 200 µM binding affinity cutoff for both VPS4 MIT domains, and the only case in which weaker, early-acting Type 2 interactors CHMP6 and CHMP4 made our binding cutoff was in the VPS4A MIT-CHMP6 complex. Nevertheless, there is good evidence that interactions between CHMP4 (Snf7 in yeast) and VPS4 are biologically important (*Kieffer et al., 2008*, *Stuchell-Brereton et al., 2007*; *Shestakova et al., 2010*; *Buysse et al., 2020*). The first of the tandem LIP5 MIT domains similarly binds ESCRT-III tails promiscuously, and again favors late-acting ESCRT-III subunits (*Yang et al., 2012*), whereas the second LIP5 MIT domain binds specifically and with high affinity to CHMP5 (*Skalicky et al., 2012*; *Yang et al., 2012*). Thus, these MIT interactions can collectively explain how $(VPS4)_6(LIP5)_{12}$ supercomplexes can bind and remodel ESCRT-III filaments that contain essentially any combination of the 12 different human ESCRT-III subunits. Moreover, their different binding affinities can explain why VPS4 remodeling activity increases as ESCRT-III filaments mature and accumulate increasing fractions of the late-acting, and higher affinity CHMP2, CHMP1, and IST1 subunits (*Pfitzner et al., 2020*).

In a similar fashion, ESCRT-III subunit binding specificities and affinities are also likely tuned to recruit other MIT cofactors as needed. It is therefore of interest that IST1, CHMP3, and CHMP1 were the most promiscuous ESCRT-III binders in our screen because each of these ESCRT-III subunits functions at transitional stages of ESCRT-III polymer maturation (*Pfitzner et al., 2020*). Our binding data suggest that many MIT proteins will bind best to mature, IST1-containing filaments that form late, when membrane constriction is greatest and fission is most imminent. In contrast, MIT proteins like AMSH and MITD1 bind many different ESCRT-III subunits with similar affinities, perhaps because deubiquitination (AMSH) and midbody stabilization (MITD1) are required throughout abscission and other ESCRT-dependent processes.

## IST1 as a versatile hub for cytokinetic cofactor recruitment

IST1 is the most promiscuous MIT binding ESCRT-III protein, and it binds most targets with high affinity (*Figure 2*, *Figure 3C*). IST1 promiscuity is achieved through combined use of two different MIM elements, and through multi-modal binding by the IST1 MIM1/3 element (*Bajorek et al., 2009a*, *Guo and Xu, 2015*; *Figure 3*). Our screen revealed that both IST1 MIM elements can mediate MIT-binding, and that they often collaborate to enhance binding (*Figure 3C*). The SPASTIN MIT-IST1

structure shows that the IST1 MIM$_{344-366}$ element can make Type 3 MIT interactions, as seen previously in the SPARTIN-MIT-IST1 complex (*Yang et al., 2008*; *Guo and Xu, 2015*). However, the same hydrophobic surface of the IST1 MIM$_{344-366}$ helix can also form Type 1 interactions with the MIT domains of ULK3, VPS4B, and LIP5 (*Guo and Xu, 2015*; *Skalicky et al., 2012*; *Caballe et al., 2015*). IST1 depletion induces severe cytokinetic defects (*Bajorek et al., 2009a*, *Agromayor et al., 2009*; *Figure 5*), reflecting its central role in abscission and as a versatile hub for MIT cofactor recruitment.

Comparison of the SPASTIN MIT complexes with IST1 and CHMP1B allowed us to design a mutation that specifically permitted only IST1 binding (*Figure 4*). This SPASTIN mutant still localized to intercellular bridges (*Figure 6A,B*) demonstrating that IST1 binding is sufficient (and CHMP1B and CHMP3 are dispensable) for midbody recruitment. Others have reported that SPASTIN midbody localization is impaired by CHMP1B depletion (*Yang et al., 2008*), but this observation can be reconciled because IST1 recruitment is also likely affected by the absence of CHMP1B (*Dimaano et al., 2008*; *Rue et al., 2008*; *Bajorek et al., 2009b*, *Goliand et al., 2018*).

## MIT binding modes of CHMP1 and CHMP3 proteins

CHMP1B and CHMP3 are also promiscuous MIT binders. CHMP1B binds two MIT domains (SPASTIN, USP8) that the closely related CHMP1A protein does not. This expanded binding range apparently reflects the ability of CHMP1B to access the Type 3 interaction mode. The Type 3 interaction extends the N-terminus of the CHMP1B MIM helix by three additional turns (vs. its Type 1 interactions). Additional interactions with this extended helix could allow SPASTIN to discriminate between CHMP1B and CHMP1A, whose sequences are highly similar at the C-terminus but diverge N-terminally (*Yang et al., 2008*). The USP8 MIT domain can also discriminate between CHMP1B and CHMP1A (*Figure 2*), and we therefore speculate that USP8 MIT may also form a Type 3 interaction with CHMP1B.

The CHMP3 MIM appears to be another example of a promiscuous helical MIM element that can bind in at least two separate grooves of different MIT domains. The CHMP3 MIM element binds the MIT domains of AMSH using a Type 4 mode (*Figure 1C*; *Solomons et al., 2011*), and LIP5 (MIT1 using a Type1/4 mode; *Skalicky et al., 2012*). Remarkably, our study suggests that CHMP3 can also adopt yet another binding mode (Type 3), as we mapped binding to the H1/H3 grooves of SPASTIN and KATNA1 MIT. The ability to adopt multiple different binding modes increases the number of possible partnerships with MIT domain proteins, and helps to explain why IST1, CHMP3 and CHMP1B are the most promiscuous MIT binders.

## SPASTIN and KATNA1 midbody recruitment and microtubule severing

ESCRT-III proteins coordinate the recruitment of the microtubule severing enzymes KATNA1 and SPASTIN to complete abscission. Following anaphase, densely packed midbody microtubules must be cleared from the midbody to facilitate cytokinetic membrane abscission (*Sharp and Ross, 2012*). Both SPASTIN and KATNA1 form hexameric rings (*Hartman and Vale, 1999*; *Eckert et al., 2012*) that can engage the C-terminal tails of TUBULIN subunits within the central pore (*Kuo and Howard, 2021*). ATP hydrolysis then drives polypeptide translocation, thereby promoting subunit exchange and/or microtubule severing (*Roll-Mecak and Vale, 2008*; *Zehr et al., 2017*). We observed distinct but overlapping midbody localization patterns for SPASTIN and KATNA1 (*Figure 5*), which may reflect their recruitment by similar ESCRT-III proteins (CHMP1B, IST1, and CHMP3) and could allow the two enzymes to act on different pools of midbody microtubules.

Our discovery that SPASTIN, but not necessarily KATNA1, is required for abscission checkpoint maintenance further underscores that these enzymes likely mediate different aspects of microtubule dynamics during abscission. Furthermore, the surprising checkpoint requirement for SPASTIN suggests that microtubule severing may also be required to support abscission arrest (in addition to physically removing microtubule barriers to allow abscission). Possible roles for SPASTIN in maintaining abscission arrest include: (1) slowing microtubule catastrophe via an ATP-independent activity (*Kuo et al., 2019*) and thereby paradoxically stabilizing severed midbody microtubules, or (2) creating shorter microtubules that promote regrowth and dynamically increase microtubule networks (*Kuo et al., 2019*; *Vemu et al., 2018*), and could thereby stabilize the midbody and/or traffic abscission factors into or out of the abscission zone (*Frémont and Echard, 2018*). Our findings point to the need for more detailed studies of the mechanistic roles of microtubule remodeling in abscission and in checkpoint regulation.

The discovery that CHMP3 and KATNB1 (*Rezabkova et al., 2017*) share overlapping binding sites on the KATNA1 MIT domain has important functional implications (*Figure 6—figure supplement 2*). KATNB1 targets KATNA1 to centrosomes during mitosis (*Matsuo et al., 2013*), whereas CHMP3 targets KATN1A to midbody arms during cytokinesis (*Figure 6*). Thus, KATNA1 likely switches partnerships from KATNB1 to CHMP3 during this cell cycle transition. KATNB1 binding also regulates KATNA1 microtubule severing activities (*Faltova et al., 2019*), specificities (*Faltova et al., 2019*), and binding partners (*Jiang et al., 2017*; *Jiang et al., 2018*), and it will therefore be of interest to learn how CHMP3 replacement alters these KATNA1 activities.

Finally, a CHMP3 mutation, T173I, located within the MIM used in our binding studies, is associated with spastic paraplegia, a disease associated with SPASTIN defects (*Cohen-Barak et al., 2022*). Our identification of CHMP3 as a SPASTIN binding partner suggests that CHMP3 may play a direct role in supporting neuronal SPASTIN functions.

## CAPN7 functions in abscission and the abscission checkpoint

Finally, we have discovered that the cysteine protease, CAPN7, localizes to midbodies and supports abscission and the abscission checkpoint. Our screen identified IST1 as the sole ESCRT-III binding partner for the CAPN7 MIT domains (*Figure 2*, *Figure 3*), in good agreement with previous reports of IST1 binding in pulldown assays (*Osako et al., 2010*; *Maemoto et al., 2011*). The CAPN7 MIT domain also reportedly binds the second alpha-helical region of the CHMP1B core domain (but not the C-terminal MIM region) (*Maemoto et al., 2011*). We did not survey this interaction because our screen included only C-terminal ESCRT-III tails, but our mutational and localization analyses indicate that CAPN7 midbody recruitment is dependent on IST1 binding (*Figure 4*, *Figure 6*).

IST1 recruitment may also enhance CAPN7 proteolytic activity within the midbody as IST1 binding was shown to activate CAPN7 proteolysis of an artificial substrate (*Osako et al., 2010*; *Maemoto et al., 2013*). CAPN7 orthologues in Aspergillus (PalB) and budding yeast (Rim13) function together with ESCRT-III binding partners to cleave the PEST peptide sequences of the transcription factors Rim101 and PacC, and thereby enhance gene expression (*Rodríguez-Galán et al., 2009*; *Subramanian et al., 2012*). Similarly, human CAPN7 reportedly binds and cleaves PEST elements of the HOXA10 transcription factor (*Yan et al., 2018*). Thus, IST1 recruitment of CAPN7 to the midbody could provide spatial and temporal control of proteolysis, leading to downstream signaling activities required for checkpoint maintenance and/or abscission.

## Cofactors for other ESCRT-dependent processes

Although we have focused on cytokinetic abscission, the same experimental framework can be used to characterize MIT cofactors for other cellular ESCRT functions, including intralumenal vesicle formation at the multi-vesicular body, nuclear envelope resealing, plasma membrane repair, and enveloped virus budding (*Christ et al., 2017*; *Vietri et al., 2020*; *Zhen et al., 2021*). Many of the newly discovered partnerships merit investigation in these other ESCRT functions. For example, CHMP7 is a specialized ESCRT-III protein that functions in post-mitotic closure and repair of the nuclear envelope (*Vietri et al., 2015*; *Gu et al., 2017*; *Olmos et al., 2015*; *von Appen et al., 2020*; *Denais et al., 2016*; *Thaller et al., 2019*). Our observation that CHMP7 interacts exclusively with the MIT domain of the AMSH deubiquitinase supports the possibility that these events may be dynamically regulated by ubiquitin-dependent processes, as has been recently reported (*Wallis et al., 2021*). More generally, our quantitative definition of the ESCRT-III-MIT interactome should provide a basis for probing how disruption of ESCRT-III and MIT cofactor activities can contribute to disease states such as hereditary spastic paraplegia (*Ciccarelli et al., 2003*), or can be used therapeutically, for example in anti-cancer strategies based on VPS4 synthetic lethality (*Neggers et al., 2020*; *Szymańska et al., 2020*).

## Materials and methods

### Key resources table

| Reagent type (species) or resource | Designation | Source or reference | Identifiers | Additional information |
|---|---|---|---|---|
| Cell line (*Homo sapiens*) | Hela-N | Maureen Powers Lab | | HeLa cells selected for transfectability |

*Continued on next page*

*Continued*

| Reagent type (species) or resource | Designation | Source or reference | Identifiers | Additional information |
|---|---|---|---|---|
| Cell line (*Homo sapiens*) | HEK293T | ATCC | CRL-3216 | |
| Antibody | Anti-CAPN7 (Rabbit polyclonal) | Proteintech | Cat#26985–1-AP | IF (1:1000) WB (1:5000) |
| Antibody | Anti-CEP55 (Sheep polyclonal) | Bastos and Barr, 2010 | | IF (1:3500) |
| Antibody | Anti-IST1 (Rabbit polyclonal) | Sundquist Lab/Covance | UT560 | WB (1:1000) |
| Antibody | Anti-CHMP4C (Rabbit polyclonal) | *Sadler et al., 2018* | | WB (1:500) |
| Antibody | Anti-KATNA1 (Rabbit polyclonal) | Proteintech | 17560–1-AP | IF (1:1000) |
| Antibody | Anti-KATNA1 (Rabbit polyclonal) | Abcam | ab111881 | IF (1:500) WB (1:1000) |
| Antibody | Anti-SPASTIN (Mouse monoclonal) | Sigma | S7074 | IF (1:1000) WB (1:1000) |
| Antibody | Anti-NUP153 (SA1) (Mouse monoclonal) | Brian Burke | | WB (1:50) |
| Antibody | Anti-NUP50 (Rabbit polyclonal) | *Mackay et al., 2010* | | WB (1:2500) |
| Sequence-based reagent | siNT | *Mackay et al., 2010* | siRNA | GCAAAUCUCCGAUCGUAGA |
| Sequence-based reagent | siCHMP4C | *Strohacker et al., 2021* | siRNA | CACUCAGAUUGAUGGCACA |
| Sequence-based reagent | siIST1 | *Bajorek et al., 2009a* | siRNA | AGAUACCUGAUUGAAAUUG |
| Sequence-based reagent | siNUP153 | *Mackay et al., 2010* | siRNA | GGACUUGUUAGAUCUAGUU |
| Sequence-based reagent | siNUP50 | *Mackay et al., 2010* | siRNA | GGAGGACGCUUUUCUGGAU |
| Sequence-based reagent | siCAPN7 | This Paper | siRNA | GCACCCAUACCUUUACAUU |
| Sequence-based reagent | siCAPN7-b | This Paper | siRNA | GGCCGUUACUGAUUGAGCU |
| Sequence-based reagent | siKATNA1 | This Paper | siRNA | GGACAGCACUCCCUUGAAA |
| Sequence-based reagent | siKATNA1-b | Horizon Discovery | CAT# L-005157 | ON-TARGET-PLUS siRNA-SMARTPOOL |
| Sequence-based reagent | siSPAS | This Paper | siRNA | GAACAGUGUGAAAGAGCUA |
| Sequence-based reagent | siSPAS-b | This Paper | siRNA | CGUUAUUGAUACUUGGAUA |
| Chemical compound, drug | Thymidine | CalBiochem | CAS 50-89-5 | 2 mM |
| Chemical compound | Oregon Green 488 maleimide | Life Technologies/Molecular Probes | O6034 | Fluorescent label for peptides |
| Software, algorithm | Fiji | NIH | RRID:SCR_002285 | |
| Software, algorithm | KaleidaGraph | Synergy Software | | |

## Identification and cloning of human MIT domains

MIT domains were selected from the literature (*Ciccarelli et al., 2003*; *Rigden et al., 2009*; *Row et al., 2007*; *Skalicky et al., 2012*; *Xiao et al., 2008*; *Fujioka et al., 2014*). Bacterial expression constructs were designed using previous reports (when available) (*Stuchell-Brereton et al., 2007*; *Caballe et al., 2015*; *Fujioka et al., 2014*; *Hadders et al., 2012*; *Solomons et al., 2011*; *Avvakumov et al., 2006*; *Yang et al., 2008*; *Guo and Xu, 2015*; *Iwaya et al., 2010*; *Yorikawa et al., 2008*; *Osako et al., 2010*; *Skalicky et al., 2012*) (NRBF2; PDB 2CRB; unpublished) or were guided by secondary structure predictions using Phyre2 (*Kelley et al., 2015*).

Bacterial expression constructs for ULK3, VPS4A, VPS4B, and LIP5 have been reported (*Caballe et al., 2015*; *Scott et al., 2005*; *Stuchell-Brereton et al., 2007*; *Skalicky et al., 2012*). Other DNAs were obtained from Addgene (RPS6KC1: 23460; ULK1: 31961), Dharmacon (SPARTIN: MHS6278-2028092; Clone ID: 5313379) and DNASU (NRBF2: HSCD00424935; KATNA1:HSCD00445516; KATN-L1:HSCD00079435; KATNAL2: HSCD00733526; KATANA1B: HSCD00042784; RPS6K-L1:HSCD00045064; AMSH: HSCD00078710; AMSHLP: HSCD00438568; USP8: HSCD004366965; SNX15: HSCD00404630; CAPN7: HSCD00404981; VPS9D1:HSCD00620221; MITD1: HSCD00356996; NRBF2:HSCD00434935). Bacterial and mammalian expression constructs for CAPN7 were further mutated to match the reference sequence NP055111.1 by making the following substitutions using quick change mutagenesis: G151S, E173V, E495K. SPASTIN was subcloned from EST ACCT 7491861 (*Han et al., 2020*). USP54 MIT was made as a gene string (Thermofisher), and the Myc-ATG13 expression construct was a gift from Do-Hyung Kim (Addgene plasmid #31965; http://n2t.net/addgene: 31965; RRID: Addgene_31965) (*Jung et al., 2009*).

Expression inserts for cell lines were generated by PCR amplification and ligated into the pLVX-tight-Puro vector (Clontech) using the NEB HiFi DNA Assembly Kit (New England Biolabs) according to the manufacturer's instructions. Human SPASTIN has four major isoforms (*Mancuso and Rugarli, 2008*; *Claudiani et al., 2005*), and we used the most abundant (M87) isoform, including residues encoded by exon 4.

A slightly longer SPASTIN MIT construct (residues 108–200) bound five-fold more tightly to CHMP1B and IST1 (CHMP1B, $K_D$ = 2.3 µM; *Figure 2*) compared to the minimal MIT sequence (residues 112–196; CHMP1B $K_D$ ~12–15 µM; data not shown and *Yang et al., 2008*) thus we used the longer construct for binding measurements. Using this construct, we were able to detect binding between SPASTIN and CHMP3; a previously unknown interaction.

Gene names, DNA sources, and amino acid sequences are given in *Supplementary file 1* and *Supplementary file 2*. All plasmids generated by this study have been deposited to Addgene for distribution (See *Supplementary file 2A* for Addgene accession numbers).

## Bacterial expression of MIT domains

Proteins were expressed in BL21 RIPL cells grown in ZYP-5052 autoinduction media (*Studier, 2005*). Transformed cells were initially grown for 3–6 hr at 37 °C, and then switched to 19 °C for an additional 20 hr. Cells were harvested by centrifugation at 5,400 x g and cell pellets were stored at –80 °C. MIT domains from VPS4A, VPS4B, ULK3, and LIP5 were expressed and purified as described previously (*Caballe et al., 2015*; *Skalicky et al., 2012*; *Scott et al., 2005*; *Stuchell-Brereton et al., 2007*).

## Purification of (His)$_6$-fusion proteins

All steps were carried out at 4 °C except where noted. Frozen cell pellets were thawed and resuspended in lysis buffer: 50 mM Tris pH 8.0, 500 mM NaCl, 1 mM Dithioreitol (DTT), 0.5 mM EDTA supplemented with 0.125% sodium deoxycholate, lysozyme (25 µg/mL) PMSF (35 µg/mL), pepstatin (1 µg/mL), leupeptin (0.5 µg/mL), aprotinin (0.1 µg/mL), DNAse1 (25 µg/mL), and 1 mM MgSO$_4$. Cells were lysed by sonication and lysates were clarified by centrifugation at 37,000 x g for 60 min. The clarified supernatant was filtered through a 0.45 µM cartridge filter and incubated with 10 mL of cOmplete His-Tag purification beads (Roche, Germany) for 45 min. Beads were washed with 500 mL of wash buffer: 25 mM Tris (pH 8.0), 500 mM NaCl, 1 mM DTT, 0.5 mM EDTA, and then with 500 mL wash buffer with 200 mM NaCl. Fusion proteins were eluted with 50 mL of 200 mM NaCl wash buffer supplemented with 200 mM imidazole, pH 8.0. Eluted protein solutions were treated with 100 µg of protease ((His)$_6$-ULP1 for (His)$_6$-SUMO fusions; GST-HRV3C preScission protease for (His)$_6$-GST fusions) in 3.5 kDa cutoff dialysis bags while dialyzing against 2x2 L of 200 mM NaCl wash buffer for 16–24 h. Uncut (His)$_6$-SUMO-MIT fusion, (His)$_6$-SUMO tag, and (His)$_6$-ULP1 proteases were removed with 5 mL of cOmplete His-Tag purification beads, and the MIT fusion proteins were concentrated and prepared for gel filtration chromatography.

For (His)$_6$ fusions of RPS6KL1, RPS6KC1, AMSH, AMSHLP, and USP8, NRBF2, MITD1, and SPARTIN, nickel column eluates were dialyzed against 2x2 L of 25 mM Tris pH 8.0 (25 °C), 50 mM NaCl, 1 mM EDTA, 1 mM DTT in the presence of ~100 µg protease ((His)$_6$ULP1 for (His)$_6$SUMO fusions; GST-HRV3C preScission protease for (His)$_6$GST fusions). Dialyzed cleavage reactions were purified by chromatography on a 5 mL HiTrapQ Sepharose Column (GE Healthcare Life Sciences, USA) (or for SPARTIN, a

HiTrap SP column) and eluted with gradient of 50–1000 mM NaCl to separate HIS-SUMO, or HIS-GST from pure MIT fractions. Pure MIT fractions were pooled, concentrated, and further purified by gel filtration chromatography.

Purification protocols for the CAPN7, SNX15 and VSP9D1 MIT domains were sufficiently different to merit separate descriptions (below).

$(His)_6$-GST-CAPN7$(MIT)_2$ was bound to GST-sepharose beads (10 mL,GE Healthcare Life Sciences, USA, 6 hr), washed with 1 L wash buffer: 25 mM Tris (pH 8), 500 mM NaCl, 1 mM DTT, 0.5 mM EDTA, and eluted with ~50 mL of wash buffer supplemented with 20 mM reduced L-glutathione (pH 8). The fusion tag was removed using ~100 µg GST-HRV3C (preScission) protease in a 3.5 kDa cutoff dialysis bag while dialyzing against 2x2 L wash buffer and 2 L of wash buffer with low salt (50 mM NaCl) over two days. The dialysate was purified by Q Sepharose chromatography (50 mL; GE Healthcare Life Sciences, USA) with a linear gradient of 50–500 mM NaCl. Fractions containing highly pure CAPN7$(MIT)_2$ were pooled and concentrated for gel filtration chromatography.

$(His)_6$-GST-SNX15 MIT was bound to cOmplete His-Tag purification beads (5 mL, Roche, Germany, 2 hr) and washed with 2 L wash buffer: 25 mM Tris (pH 8), 500 mM NaCl, 1 mM DTT, 0.5 mM EDTA, and eluted with wash buffer supplemented with 250 mM imidazole (pH 8). The eluent was bound to glutathione sepharose beads (10 mL, GE Healthcare Life Sciences, USA, 6 hr), washed with 1 L wash buffer: 25 mM Tris (pH 8), 500 mM NaCl, 1 mM DTT, 0.5 mM EDTA, and eluted with ~50 mL of wash buffer supplemented with 20 mM reduced L-glutathione (pH 8). The fusion tag was removed using ~100 µg GST-HRV3C (preScission) protease in a 3.5 kDa cutoff dialysis bag while dialyzing against 2x2 L wash buffer and 2x2 L of wash buffer with low salt (25 mM NaCl) over 2 days. The dialysate was purified by Q Sepharose chromatography (50 mL, GE Healthcare Life Sciences, USA). SNX15 MIT eluted in the flow through and was concentrated for gel filtration chromatography.

$(His)_6$-GST-VPS9D1 MIT was lysed in 50 mM Tris pH 7.4 (25 °C), 500 mM NaCl, 0.5 mM EDTA, and 1 mM TCEP supplemented with 0.125% sodium deoxycholate, lysozyme, PMSF, pepstatin, leupeptin, aprotinin and DNAse I (as described above). The clarified cell lysate was bound to 10 mL of cOmplete His-Tag purification resin (Roche, Germany, 30 min), washed with 1 L base buffer, and eluted with ~50 mL of base buffer supplemented with 250 mM imidazole (pH 8.0). The fusion tag was removed by incubation with ~100 µg GST-HRV3C protease in 3.5 kDa cutoff dialysis bag while dialyzing against 2x2 L of 25 mM Tris pH 7.4 (25 °C), 50 mM NaCl, 1 mM DTT, 0.5 mM EDTA. The dialysate was purified by Q Sepharose chromatography (50 mL, GE Healthcare Life Sciences, USA) and eluted with a 50–1000 mM NaCl linear gradient. VPS9D1 eluted in the flow through and was concentrated for gel filtration chromatography.

Finally, all MIT proteins were purified by Superdex 75 gel filtration chromatography (GE Healthcare Life Sciences, USA) in 25 mM Tris (pH 7.2 at 25 °C), 150 mM NaCl, 1 mM DTT, and 0.5 mM EDTA. MIT protein fractions were pooled and concentrated. MIT domain masses were confirmed with ESI-MS or MALDI-TOF (University of Utah Mass Spectrometry Core Facility, see *Supplementary file 1A*). Yields ranged between 0.5–35 mg/L of bacterial culture.

## Expression and purification of ESCRT-III C-terminal peptides

ESCRT-III C-terminal peptides were prepared as described previously (*Talledge et al., 2018*). Briefly, most peptides were expressed as $(His)_6$-SUMO-fusions, except for IST1-MIM$_{316-343}$ and IST1 MIM$_{344-366}$ which were made synthetically by the University of Utah Peptide Synthesis Core. Peptides were expressed in BL21-Codon Plus (DE3) RIPL cells (Agilent, Santa Clara, CA, USA) in ZYP-5052 auto-induction media (*Studier, 2005*). Cells were lysed by sonication in lysis buffer (40 mL/L of culture) containing 50 mM Tris, pH 7.2, 150 mM NaCl, 5 mM imidazole, 2 mM DTT, 0.5 mM EDTA, and 0.125% sodium deoxycholate, supplemented with lysozyme, protease inhibitors, and DNAse I (Roche, Germany) (as described above). Clarified cell lysates were incubated with 10 mL of cOmplete His-Tag purification resin (Roche, Germany) for 30 min, washed with 500 mL wash buffer: 50 mM Tris, pH 7.2, 500 mM NaCl, 5 mM imidazole, 5 mM DTT, 0.5 mM EDTA, and then with 500 mL wash buffer containing 150 mM NaCl. $(His)_6$-SUMO affinity tags were removed by on-column cleavage with $(His)_6$-ULP1 (0.7 mg, overnight, 4 °C) in 40 mL of the 150 mM NaCl wash buffer. Cleaved peptides were collected from the column flow through and dialyzed against 25 mM sodium phosphate, pH 6.5, 50 mM NaCl, 2 mM DTT, 0.5 mM EDTA, and then purified by Q-sepharose ion exchange chromatography (GE Healthcare Life Sciences, USA) with a linear gradient from 50 mM to 1 M NaCl. Peptide fractions were pooled and

dialyzed against 25 mM Tris, pH 7.2, 150 mM NaCl, 1 mM TCEP, 0.5 mM EDTA, and further purified by Superdex-75 size exclusion chromatography (GE Healthcare Life Sciences, USA). Typical IST1 peptide yields were 4.5 mg/L culture. Purified ESCRT-III C-terminal fragments contained non-native 'GlyCys' or 'Cys' residues at their N-termini, and masses were confirmed by mass spectrometry (see *Supplementary file 1B*) either before labeling (CHMP4C, CHMP1A$_{140-196}$, CHMP1B$_{143-199}$) or after labeling (all other peptides; dye adds a mass shift of 463.4 Da). Competition experiments used peptides lacking N-terminal 'Cys' residues to avoid disulfide cross-linking (see *Supplementary file 1C*).

## Peptide fluorescent labeling

Fluorescent labeling was performed by the University of Utah DNA/Peptide Synthesis Core as described previously (*Caballe et al., 2015*; *Talledge et al., 2018*). Briefly, peptides were labeled in DMSO using ~1.3-fold molar excess of Oregon Green 488 maleimide (Life Technologies/Molecular Probes #O6034, USA) dissolved in a 1:1 solution of acetonitrile:DMSO. Reversed phase HPLC was used to monitor the reactions and separate labeled peptides from unreacted dye and unlabeled peptides using reversed phase HPLC. Labeled peptide fractions were dried under vacuum and dissolved in water. Peptide concentrations were quantified using the absorbance of Oregon Green 488 at 491 nm (e=83,000 cm$^{-1}$ M$^{-1}$ in 50 mM potassium phosphate, pH 9.0).

## Fluorescence polarization anisotropy binding assays

Binding experiments were performed as described previously (*Caballe et al., 2015*) in 25 mM Tris, pH 7.2, 150 mM NaCl, 0.1 mg/mL Bovine Serum Albumin (BSA), 0.01% Tween-20, and 1 mM DTT, with 250–500 pM fluor-labeled ESCRT-III peptides and purified MIT domains at the indicated concentrations. A Biotek Synergy Neo Multi-Mode plate reader (Biotek, USA) was used to measure fluorescence polarization with excitation at 485 nm and emission (detection) at 535 nm. Binding isotherms were fit to 1:1 models using KaleidaGraph (Synergy Software) as described previously (*Skalicky et al., 2012*; *Talledge et al., 2018*). Reported K$_D$ values are averages from at least three independent isotherms. Non-binding (K$_D$ >200 μM) was confirmed at least twice independently. The interactions between IST1 and SPASTIN as well as IST1 and USP8 failed to generate the large changes in polarization needed to reliably fit dissociation constants. We therefore used a C-terminal labeled IST1 peptide to measure binding with SPASTIN, and competitive binding experiments were used to calculate the K$_i$ for the interaction of USP8 with IST1 (*Figure 2—figure supplement 5E*). Competition experiments were performed as described previously (*Sadler et al., 2018*). Briefly, complexes of MIT domains and fluorescently labeled peptides (26.6 μM MITD1 and 0.5 nM CHMP4B peptide; 0.75 μM USP8 and 0.5 nM IST1 peptide, 12.5 μM USP8 MIT and 0.5 nM CHMP4B peptide) were titrated with the indicated concentrations of unlabeled peptides. IC$_{50}$s were calculated using KaleidaGraph (Synergy Software) and then converted to K$_i$ values (*Cer et al., 2009*). Competitive binding curves were measured independently three or more times for each peptide, and one to three times for each positive control peptide.

## Co-immunoprecipitation experiments

### KATNA1 pulldowns

HEK293T cells were seeded at 0.5x10$^6$ cells per well in six-well plates and transfected 24 hr later with 1 μg of plasmid encoding Myc-KATNB1 and one of the following: 2 μg empty vector control, 1.5 μg OSF-KATNA1-WT, 2 μg of OSF-KATNA1-R14A, or 1.5 μg of OSF-KATNA1-V55D P60 using PEI (10 ul per well of 1 mg/mL). Empty vector was added as necessary to bring the total transfected DNA to 3 μg/well. Cells were harvested 48 hr post-transfection and lysed in 400 μl of 50 mM Tris, pH 7.2, 150 mM NaCl, 0.5% Triton-X100, 1 mM DTT, protease inhibitors (1:100, Sigma). Lysates were clarified by centrifugation at 16,100 x g for 10 min at 4 °C, and clarified lysates were incubated with 20 μl of a 50% slurry of Strep-Tactin resin (IBA Biosciences) for 30 min. Beads were washed 4 x with 500 μl lysis buffer. After the final wash, Strep-Tactin beads were aspirated to near dryness and bound proteins were eluted by boiling in 40 μl of 2 x Laemmli sample buffer, resolved by SDS-PAGE, and detected by Western blotting.

### ULK(MIT)$_2$ pulldowns

HEK293T cells were seeded in 6 well plates and transfected using PEI as above and DNA levels were optimized to normalize expression levels: 3 μg pCAG-OSF-ULK3(MIT)$_2$ (residues 277–449) and 250 ng

of pCAG-CHMP1A-myc, pCAG-CHMP1B-myc, or pCAG-Myc-IST1; 1.5 µg pCAG-OSF-PP-ULK1(MIT)$_2$ (residues 833–1050) and 500 ng of pCAG-CHMP1A-myc, pCAG-CHMP1B-myc, or pCAG-Myc-IST1; 500 ng pCAG-OSF-PP-ULK1(MIT)$_2$ and 2 µg Myc-ATG13; 2.5 µg pCAG-OSF empty vector and 500 ng of pCAG-CHMP1A-myc, pCAG-CHMP1B-myc, or pCAG-Myc-IST1; 1 µg pCAG-OSF empty vector and 2 µg Myc-ATG13. Cells were harvested 24 hr post transfection and lysed in 50 mM Tris, pH 7.2, 1% Triton, 150 mM NaCl, 1 mM DTT, 1:100 mammalian protease inhibitors (Sigma). Lysates were clarified by centrifugation at 16,100 x g for 10 min at 4 °C, and incubated with 20 µl of a 50% slurry of Strep-Tactin resin (IBA Biosciences) for 30 min at 4 °C. Beads were washed 4 x with 500 µl lysis buffer. After the final wash, Strep-Tactin beads were aspirated to near dryness and bound proteins were eluted and detected by Western blotting as described above.

For ULK(MIT)$_2$ pulldowns with CHMP2A (*Figure 2—figure supplement 6*); 10 cm dishes of 60–80% confluent HEK293T cells were individually transfected with PEI and 12 µg of pCAG-OSF-ULK3(MIT)$_2$, pCAG-OSF-PP-ULK1(MIT)$_2$, pCAG-CHMP2A-myc, pCAG-CHMP2A-L216D/L219D, Myc-ATG13, or pCAG-OSF-empty vector control. Cells were harvested 24 hr post-transfection, lysed in 25 mM Tris, pH 7.2, 150 mM NaCl, 1 mM DTT, 1% Triton-X100 supplemented with 1:100 mammalian protease inhibitors (Sigma), and lysates were clarified by centrifugation at 16,100 x g for 10 min at 4 °C. Clarified lysates expressing CHMP2A-Myc were mixed with lysates expressing OSF-ULK(MIT)$_2$ or empty vector controls and incubated overnight at 4 °C. Lysate mixes were then incubated with 20 µl of a 50% slurry of Strep-Tactin resin (IBA Biosciences) for 1 hr at 4 °C. Beads were washed 5 x with 500 µl of 25 mM Tris, pH 7.2, 150 mM NaCl, 1 mM DTT, 0.5% Triton-X100 buffer. After the final wash, Strep-Tactin beads were aspirated to near dryness, and bound proteins were eluted and detected by Western blotting.

## X-ray crystallography

SPASTIN MIT (residues 112–196) and IST1 peptide (residues 344–366) were prepared as described above, mixed at a 1:1.2 molar ratio (14.25 mg/ml SPASTIN MIT and 4.8 mg/ml IST1 peptide), and filtered using a 0.2 µM cartridge filter. This complex was mixed in a 2:1, 1:1, and 1:2 (v/v) ratio with 100 mM sodium cacodylate/ hydrochloric acid pH 6.5, 40% (v/v) PEG 300, 200 mM calcium acetate (0.6 µL final volume; Wizard Cryo 1/2 screen (Rigaku, USA), condition D1). Crystals formed by sitting drop vapor diffusion after ten days at 4 °C in 1:1 and 1:2 (v/v) rations. A crystal from 1:1 or 1:2 ratio was suspended in a small nylon loop and cryocooled by plunging in liquid nitrogen. Supplemental cryoprotection was not used for the crystals grown in these conditions.

X-ray diffraction data were collected at the Stanford Synchrotron Radiation Lightsource (SSRL) using beamline 9.1. The crystal was maintained at 100 °K with the aid of a cold nitrogen gas stream during data collection. Data were integrated and scaled using XDS (*Kabsch, 2010a*, *Kabsch, 2010b*) and AIMLESS (*Evans and Murshudov, 2013*; *Evans, 2011*). Initial phases were obtained from phenix-phaser (*Bunkóczi et al., 2013*) using SPASTIN MIT (PDB 3EAB) (*Yang et al., 2008*) as a search model. The resulting electron density was readily interpretable and further built using Coot (*Emsley and Cowtan, 2004*; *Emsley et al., 2010*) and phenix-refine (*Liebschner et al., 2019*).

The model was evaluated using Molprobity (*Davis et al., 2007*; *Williams et al., 2018*) and judged to be of good quality. Two regions of unexplained density were present in the Fo-Fc electron density map. PEG 300 is present at 40% (v/v) in the crystallization condition and the density resembled smaller PEG molecules. Several molecules were tested by model building and refinement and the best fit to the density was obtained with one molecule of triethylene glycol (PGE) and one of tetraethylene glycol (PG4). The PEG molecules improved the model statistics and maintained good geometry. Two initially assigned water molecules also produced unexplained density in the Fo-Fc electron density map. Several ions were tested by model building and refinement until the best fit was obtained with one chloride ion and one calcium ion. Both ions are present in the crystallization condition (hydrochloric acid and calcium acetate), and their inclusion improved the model statistics and maintained good geometry. The final model refined to $R_{work}$ = 0.149 and $R_{free}$ = 0.159. Structure coordinates have been deposited in the RCSB Protein Data Bank under PDB ID 7S7J. Full statistics and data collection details are provided in *Supplementary file 3*.

An omit map of the IST1 portion of the complex was generated for figure presentation. This map was calculated by removing IST1 from the final model followed with phenix refinement and simulated annealing (Cartesian) to minimize phase bias (*Figure 4—figure supplement 1*).

## Cell culture

HEK293T and HeLa cells were cultured and maintained at 37 °C and 5% $CO_2$ in DMEM supplemented with 10% FBS. TetOn-HeLa cells were supplemented with 500 µg/mL G418 (Invitrogen) to maintain expression of the Tet-On Advanced protein. DOX-inducible cell lines generated in the parental TetOn-HeLa cell line were supplemented with 500 µg/mL G418 + 0.5 µg/mL puromycin (Invivogen).

## Cell lines

Our parental HeLa cell line was authenticated by genomic sequencing of 24 loci (University of Utah Sequencing Core) and confirmed to be mycoplasma-free by routine PCR testing (ABM) following the manufacturer's protocols. HeLa cells were transfected with the pLVX-TetOn-Advanced plasmid (Clontech) and selected with 500 µg/mL G418 for 14 days. Single colonies were isolated, expanded, and screened for TetOn-Advanced expression by western blot using a TetR monoclonal antibody. The optimal clone, which was selected based on a combination of TetOn Advanced expression and tight control of DOX-inducible expression, was used as the parental HeLa TetOn line. To generate stable cell lines with doxycycline-inducible expression, the parental TetOn cell line was transfected with pLVX-tight puro plasmids containing the MIT genes of interest (see *Supplementary file 2A*) and selected for 14 days in 500 µg/mL G418 +0.5 µg/mL puromycin. Single colonies were expanded and screened for expression by immunofluorescence and western blotting. Selected clones were further validated by sequencing the PCR amplified MIT gene of interest from genomic DNA. Protein expression was induced by addition of 1 µg/mL doxycycline.

## siRNA transfections

For experiments in *Figure 5*, transfection protocols were as follows: Day 1–350,000 cells were reverse transfected with 20 nM siRNA targeting MIT protein (as indicated) in a 35 mm dish using Lipofectamine RNAiMAX and following manufacturer's instructions; Day 2 – cells from Day 1 were trypsinized, resuspended in a total volume of 6 mL DMEM, and divided as follows: 0.5 mL into each of four wells of a 24-well dish containing 12 mm circle glass coverslips, and 2 mL into each of two 35 mm dishes. These samples were again reverse transfected with 20 nM siRNA (as indicated); for active abscission checkpoint samples, cells in two wells of a 24-well dish (for Immunofluorescence) and one 35 mm dish (for Western blot) were also reverse transfected with 10 nM siNUP153 and 10 nM siNUP50 at this time, incubated for 8 hr, then treated with 2 mM thymidine for 24 hr; Day 3 – thymidine was removed, cells were washed 2 x with warm PBS, and fresh medium added to all cells; Day 4–16 hr after thymidine release cells were harvested for analysis. For localization experiments in *Figure 6*, protein expression was induced by incubating cells in the presence of 1 µg/mL doxycycline for the duration of the 48 hr experiment. siRNA transfection protocols (in the presence of doxycycline) were as follows: Day 1 – either 70,000 cells (in one well of a 24-well dish containing a 12 mm glass coverslip for IF) or 350,000 cells (in a 35 mm dish for Western Blot) were reverse transfected with 20 nM siRNA targeting the indicated MIT protein plus 10 nM siNUP153 and 10 nM siNUP50 (siNups), incubated for 8 hr, then treated with 2 mM thymidine for 24 hr; Day 2 – thymidine was removed and fresh medium including 1 µg/mL doxycycline was added to all cells (as described above); Days 3–16 hr after thymidine release cells were harvested for analysis. siRNA sequences are reported in *Supplementary file 2B*.

## Immunoblotting

Cells were lysed in RIPA buffer (Thermo Fisher) supplemented with mammalian protease inhibitor cocktail (Sigma; used at 1:100 dilution) for 15 min on ice with brief vortexing every 5 min. Lysates were cleared by centrifugation at 17,000 x g for 10 min at 4 °C. Lysate protein concentrations were determined using the BCA Assay (Thermo Fisher) and normalized prior to SDS-PAGE. 12 µg lysate per sample were prepared with SDS loading buffer, resolved by SDS-PAGE, and transferred to either PVDF or nitrocellulose. Membranes were blocked for 1 hr at room temperature in 5% milk in TBS, then incubated overnight at 4 °C with primary antibodies (see *Supplementary file 2C* for dilutions). Following 3x10 min washes in TBS-T, membranes were incubated with the corresponding secondary antibodies for 1 hr at 23 °C, washed again with TBS-T, and imaged using a LiCor Odyssey infrared scanner.

## Immunofluorescence imaging and phenotype quantification

Cells were seeded on fibronectin-coated glass coverslips and treated with the indicated siRNAs, according to the individual experimental protocol as described in 'siRNA transfections'. For analysis,

cells were briefly washed with 1 X PHEM buffer (60 mM PIPES, 25 mM HEPES, pH 6.9, 1 mM EGTA, 2 mM MgCl$_2$) and then fixed for 20 min at 23 °C in 4% formaldehyde + 0.5% Triton X-100 (in 1 X PHEM buffer). Following fixation, cells were washed with PBS and incubated 30 min in blocking buffer (3% FBS +0.1% Triton X-100 in PBS). Primary antibodies were incubated for least 1 hr at 23 °C (see *Supplementary file 2C* for dilutions). Coverslips were then washed with PBS and incubated with the secondary antibodies (Thermo Fisher) for 1 hr at 23 °C. Following a final wash in PBS, coverslips were mounted onto glass microscope slides using Prolong Gold Antifade Reagent with or without DAPI (Thermo Fisher). In the case of KATNA1, localization was confirmed with two independent antibodies (Proteintech – #17560–1-AP; Abcam – ab111881) and two different fixation conditions: –20 °C Methanol for 10 min and PFT (1 x PHEM + 4% PFA +0.5% Triton as described above) (data not shown).

Images were acquired using a Nikon Ti-E inverted microscope equipped with a 60X PlanApo oil immersion objective, an Andor Zyla CMOS camera, and an automated Prior II motorized stage controlled with the Nikon Elements software. For phenotype quantification in *Figure 5*, the software was used to acquire 49 images using a randomized 7 x 7 grid pattern. The images were then blinded and scored independently by four individuals. For *Figure 6*, 25 images for each treatment were acquired, blinded, and scored independently by three individuals. Quantification and statistical analyses were performed using GraphPad Prism.

## Acknowledgements

We thank B Burke, J Martin-Serrano for antibodies, and D Ayer for the pLVX vector. We thank B Zak for assistance in image quantification, S Alam, F Whitby, and C Hill for assistance with X-ray data collection and analysis, M Saunders for bioinformatics searches not described here, A Fields, K Miller, I Torres, and E Johnston for help purifying proteins, S Endicott and M Hanson (Utah peptide synthesis Core) for labeling peptides, and S Osburn-Staker and K Parsawar (Utah mass spectrometry Core) for measuring masses of purified peptides and proteins. Microscopy data collected on the Nikon Automated Widefield microscope was performed in the University of Utah Cell Imaging Core. Oligonucleotides and some peptides were synthesized by the DNA/Peptide Facility, and sequencing was performed by the DNA sequencing Core Facility at the University of Utah. We also acknowledge biostatistics advice from the Cancer Biostatistics Shared Resource at the Huntsman Cancer Institute at the University of Utah which is supported by the National Cancer Institute of the National Institutes of Health under Award Number P30CA042014. This work was funded by grants from NIH 5R01GM112080 (WIS, KSU) and 5R01GM131052 (KSU, DM). DMW was supported in part by an American Cancer Society Postdoctoral Fellowship PF-14-102-01-CMS. ELP was supported by a predoctoral NIH fellowship F31GM139318. Use of the Stanford Synchrotron Radiation Lightsource, SLAC National Accelerator Laboratory, is supported by the U.S. Department of Energy, Offices of Science, Office of Basic Energy Sciences under Contract No. DE-AC02-76SF00515. The SSRL Structural Molecular Biology Program is supported by the DOE Office of Biological and Environmental Research, and by the NIH, National Institute of General Medicine Sciences (including P41GM103393). The contents of this publication are solely the responsibility of the authors and do not necessarily represent the official views of NIGMS or NIH. The funding sources were not involved in the study design, data collection and interpretation, or decision to submit the work for publication.

## Additional information

### Funding

| Funder | Grant reference number | Author |
| --- | --- | --- |
| National Institutes of Health | 5R01GM112080 | Katharine S Ullman Wesley I Sundquist |
| National Institutes of Health | 5R01GM131052 | Douglas R Mackay Katharine S Ullman |
| American Cancer Society | PF-14-102-01-CMS | Dawn M Wenzel |

| Funder | Grant reference number | Author |
|--------|------------------------|--------|
| National Institutes of Health | F31GM139318 | Elliott L Paine |

The funders had no role in study design, data collection and interpretation, or the decision to submit the work for publication.

## Author contributions

Dawn M Wenzel, Jack J Skalicky, Conceptualization, Formal analysis, Investigation, Methodology, Writing – original draft, Writing – review and editing; Douglas R Mackay, Conceptualization, Formal analysis, Investigation, Methodology, Writing – review and editing; Elliott L Paine, Formal analysis, Investigation, Methodology, Writing – review and editing; Matthew S Miller, Formal analysis, Investigation, Writing – review and editing; Katharine S Ullman, Wesley I Sundquist, Conceptualization, Supervision, Funding acquisition, Methodology, Project administration, Writing – review and editing

## Author ORCIDs

Dawn M Wenzel ⓘ http://orcid.org/0000-0002-3605-6220
Douglas R Mackay ⓘ http://orcid.org/0000-0002-2698-2140
Jack J Skalicky ⓘ http://orcid.org/0000-0002-5450-0567
Elliott L Paine ⓘ http://orcid.org/0000-0001-5575-297X
Katharine S Ullman ⓘ http://orcid.org/0000-0003-3693-2830
Wesley I Sundquist ⓘ http://orcid.org/0000-0001-9988-6021

## Decision letter and Author response

Decision letter https://doi.org/10.7554/eLife.77779.sa1

## Additional files

### Supplementary files

• Supplementary file 1. MIT domain proteins and ESCRT-III peptides used for the fluorescence polarization screen and X-ray crystallography. (A) MIT domain protein constructs used in our screen. (B) Fluorescently labeled ESCRT-III C-terminal tails used in the fluorescent polarization screen. (C) Unlabeled ESCRT-III C-terminal peptide tails used for competition experiments and structural biology.

• Supplementary file 2. Plasmids, siRNA and antibodies used in this study. (A) Plasmids, (B), siRNA sequences, and (C) antibodies used for this study.

• Supplementary file 3. SPASTIN MIT-IST1 complex data collection and refinement statistics.

• Transparent reporting form

### Data availability

Diffraction data have been deposited in the PDB under the accession code 7S7J. All plasmids have been deposited in the Addgene plasmid repository. Source data files have been included for Figure 2-figure supplement 6, Figure 5-figure supplement 1, Figure 6 -figure supplement 1, Figure 6- figure supplement 2.

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

## Appendix 1

### Supplemental discussion

Meiotic clade AAA-ATPases

Single N-terminal MIT domains are prominent in most meiotic clade AAA+ ATPases including VPS4A, VPS4B, SPASTIN, KATNA1, and KATNAL1 (*Monroe and Hill, 2016*). These ATPases function as hexamers and avidity will therefore enhance their binding to polymeric ESCRT-III binding partners. KATNAL2 and FIDGETIN are the only meiotic clade ATPases that appear to lack MIT domains. In the place of a MIT domain, KATNAL2 contains an LisH domain (*Iwaya et al., 2010*), which did not bind any ESCRT-III proteins in our screen (*Figure 2—figure supplement 4A*), and secondary structure predictions do not reveal three characteristic MIT alpha helices at the N-terminus of FIDGETIN (data not shown). Of the AAA+ ATPases, the VPS4A and B MIT domains are the most promiscuous ESCRT-III interactors, and previous studies using biosensor binding and yeast two hybrid approaches had identified CHMP1A, 1B, 2 A, 3, 4 A-C, 6, and IST1 as VPS4 binding partners (*von Schwedler et al., 2003*; *Scott et al., 2005*; *Stuchell-Brereton et al., 2007*; *Kieffer et al., 2008*, *Bajorek et al., 2009a*). Our screen confirmed VPS4 binding to CHMP1A, 1B, 2 A, 2B, 6, and IST1, with 1:1 $K_D$'s ranging from 0.5 to 54 µM, with VPS4B consistently showing 3–20 fold weaker binding than VPS4A. CHMP3 and the CHMP4 proteins are also likely biologically relevant VPS4 MIT binding partners (*Lin et al., 2005*; *Kieffer et al., 2008*, *Stuchell-Brereton et al., 2007*; *Adell et al., 2014*), but their 1:1 binding affinities were below our cutoff. Thus, VPS4 enzymes bind all ESCRT-III paralogs, except the specialized CHMP5 and CHMP7 subunits, consistent with the central role that VPS4 enzymes play in remodeling ESCRT-III filaments.

SPASTIN MIT, in contrast, binds specifically to three ESCRT-III proteins: CHMP1B (*Yang et al., 2008*), IST1 (*Yang et al., 2008*; *Renvoisé et al., 2010*), and CHMP3. We confirmed the previously described interactions, the newly discovered CHMP3 interaction, and the lack of interaction with the remaining nine non-binding ESCRT-III proteins ($K_D$ >200 µM). Both CHMP1B and IST1 form an amphipathic helix that binds in the MIT groove formed by helices H1 and H3 (*Yang et al., 2008*) (Type 3 binding, *Figure 4*, and *Figure 4—figure supplement 1*). The CHMP1B and IST1 binding surfaces on the MIT are overlapped, leading to mutually exclusive and competitive binding. A longer construct of SPASTIN (residues 108–200) enabled detection of a novel binding interaction with CHMP3 with the H1-H3 groove. Although we lack structural data for this interaction, our mutational analysis supports the idea that all three ESCRT-III proteins, CHMP1B, CHMP3 and IST1 compete for the same binding site (*Figure 4* and *Figure 4—figure supplement 2*). The moderate affinity of CHMP3 for SPASTIN may have precluded detection by previous studies using yeast two hybrid approaches (*Reid et al., 2005*; *Yang et al., 2008*; *Agromayor et al., 2009*).

Our screen revealed for the first time that KATNA1 and KATNAL1 MITs are also specific binders, in this case interacting with only CHMP3 (*Figure 2*). CHMP3 binding to the KATNA1 MIT domain was eliminated by the H1/3 groove mutant V55D, which is consistent with Type 3 binding (*Figure 4G*). Though KATNA1 and KATNAL1 binding constants are relatively weak, 32 and 120 µM respectively, avidity is likely prevalent for these interactions *in cell* due to the hexameric association of KATNA1/KATNAL1 subunits and polymeric CHMP3 assemblies.

Proteases

The deubiquitinases AMSH, AMSHLP, USP8 and USP54, and the protease CAPN7 all contain MIT domains that can bind ESCRT-III subunits. CAPN7 is discussed in the main text, and therefore will not be extensively discussed here, and structures of the MIT domains of USP8 (*Avvakumov et al., 2006*) and AMSH (*Solomons et al., 2011*) have been determined by others. The MIT domain from the deubiquitinase AMSH is promiscuous, binding 10/12 ESCRT-III proteins: CHMP1A, 1B, 2 A, 2B, 3, 4 A, 4B, 4 C, 5 and 7. Seven of these interactions have been reported (*Tsang et al., 2006*; *Ma et al., 2007*; *Agromayor and Martin-Serrano, 2006*; *Row et al., 2007*; *Solomons et al., 2011*), whereas the CHMP5 and CHMP7 interactions are new. The CHMP5 (*Colcher et al., 1977*; *Tsang et al., 2006*) and CHMP4C (*Tsang et al., 2006*) interactions tested negative in previous screens, and the CHMP7 interaction had not been tested previously. CHMP3 makes a tight ($K_D$ = 0.011 µM) Type 4 interaction with AMSH MIT (*Solomons et al., 2011*; *Figures 1C and 2*). The remaining ESCRT-III proteins are strong to moderate binders ($K_{D's}$ = 7–85 µM), and CHMP6 and IST1 do not bind ($K_D$ >200 µM). The AMSH paralogue, AMSHLP, also binds preferentially to CHMP3, albeit weakly. To our knowledge, this protein had not previously been tested for ESCRT-III interactions.

USP54 is a catalytically inactive deubiquitinase, and was predicted to contain an MIT domain in a bioinformatics study (*Rigden et al., 2009*). We found that the USP54 MIT domain bound weakly to all 12 ESCRT-III proteins ($K_D$ = 70–170 µM, see *Figure 2—figure supplement 3*), suggesting nonspecific binding in our assay. CHMP1B, CHMP2A, CHMP2B, CHMP4C and CHMP6 were previously identified as USP54 MIT binding partners in in vitro pulldown binding assays (*Rigden et al., 2009*). USP54 is overexpressed in colorectal cancer (*Fraile et al., 2016*) and mutations outside of the predicted MIT domain are enriched in patients relapsing with acute lymphoblastic leukemia (*Xiao et al., 2016*). Mechanistic roles for WT and mutant USP54 proteins in tumorigenesis remain to be determined.

The USP8 MIT domain bound CHMP1B, CHMP2A, CHMP2B, CHMP3, CHMP4A, CHMP4B, and IST1 in our screen (*Figure 2*, *Figure 2—figure supplement 1*). We did not observe the CHMP1A or CHMP4C binding reported in other studies (*Row et al., 2007*; *Rigden et al., 2009*), but we uniquely detected CHMP4A and CHMP4B binding (*Figure 2*, *Figure 2—figure supplement 1*, *Figure 2—figure supplement 5*). The tightest interactions were with IST1 ($K_D$ = 0.5 µM) and CHMP3 ($K_D$ = 12 µM) while the other interactions were weak ($K_D$ = 70–185 µM). IST1 binding had been reported previously (*Agromayor et al., 2009*).

## Kinases

Three MIT proteins contain kinase domains: ULK3, RPS6KC1, and RPS6KL1. Of these, only ULK3 has been shown to have catalytic activity (*Caballe et al., 2015*; *Maloverjan et al., 2010a*, *Maloverjan et al., 2010b*), and RPS6KC1 and RSP6KL1 are predicted to contain pseudo-kinase domains (*Hayashi et al., 2002*). Our screen confirmed ULK3 MIT interactions with CHMP1A, CHMP1B, CHMP2A and IST1, which were previously identified by pulldown and yeast two hybrid assays (*Caballe et al., 2015*), and we found that ULK3 (MIT)$_2$ also binds CHMP3. The MIT domains of RPS6KC1 and RPS6KL1 had not previously been assayed for ESCRT-III binding, and our new findings that they interact with IST1 and CHMP3, now link these understudied proteins to the ESCRT pathway. Interestingly, RPS6KC1 binds sphingosine kinase 1 (SPHK1) and localizes to endosomes (*Hayashi et al., 2002*; *Liu et al., 2005*), where ESCRT machinery creates cargo-carrying intralumenal (*Hurley, 2015*) and cytoplasmic vesicles (*McCullough et al., 2015*; *Allison et al., 2013*).

## Other MIT proteins

MITD1 showed the broadest ESCRT-III promiscuity of the other MIT proteins tested, binding CHMP1A, 1B, 2 A, 3, 4 A, 4B, 6 and IST1. The CHMP1A, 1B, 2 A, and IST1 interactions were detected previously using pulldowns and yeast two hybrid assays (*Hadders et al., 2012*; *Lee et al., 2012*; *Agromayor et al., 2009*), and a structure of the MITD1 MIT-CHMP1A complex has been reported (*Hadders et al., 2012*). Interestingly, in their structure, Hadders and colleagues identified a PEG molecule bound to the H1/H3 groove of one of two inequivalent MITD1 MIT domains and noted that this might indicate a propensity for binding ESCRT-III molecules in this groove (*Hadders et al., 2012*). The new interactions that we identified with CHMP3, CHMP4A, CHMP4B, and CHMP6 could, in principle, all utilize the H1/H3 groove because CHMP4A, 4B, and 6 all make Type 2 interactions (*Kieffer et al., 2008*, *Samson et al., 2008*), and CHMP3 can apparently bind in this groove in a Type 3 mode (see main text for details). A key difference between our experiments and those reported previously is that our screen employed isolated MIT domains whereas others have used full-length proteins. In the full-length protein, the H1/H3 groove may be autoinhibited by the C-terminal PLD domain (*Figure 1B*) and may therefore only be accessible in specific biological contexts, such as when the protein binds membranes or dimerizes. As noted in the main text, MITD1 stabilizes the intercellular bridge to promote proper abscission (*Hadders et al., 2012*; *Lee et al., 2012*), and this extended function may explain the need for promiscuous ESCRT-III binding.

LIP5 was also quite promiscuous, and bound CHMP1A, 1B, 2 A, 2B, 3, 5, and IST1. We, and others had previously reported these interactions (*Shim et al., 2008*; *Ward et al., 2005*; *Skalicky et al., 2012*; *Guo and Xu, 2015*). In contrast to Guo et al (*Guo and Xu, 2015*), however, we found that the IST1 MIM$_{316-343}$ element contributed to binding, as an IST1 peptide that encompassed both MIM elements bound ~10 fold more tightly than a peptide spanning MIM$_{344-366}$ alone (*Figure 3C*). As mentioned in the main text, the LIP5(MIT)$_2$-CHMP5 complex was the highest affinity interaction seen in all of our studies (2.5 nM), which is explained by the Type 5 interaction mode, in which the extended CHMP5 tail and associated linkers wrap almost completely around the second MIT domain (*Skalicky et al., 2012*; *Yang et al., 2012*; see *Figure 1C*). In contrast, other ESCRT-III proteins bind

in the H2/H3 groove of the first MIT domain in a Type 1 interaction (*Guo and Xu, 2015*; *Skalicky et al., 2012*).

The MIT domains of SNX15 and SPARTIN both bind exclusively to IST1. The SPARTIN-IST1 interaction had been reported previously (*Renvoisé et al., 2010*; *Guo and Xu, 2015*), and we found that the IST1 MIM$_{344-366}$ element provided most of the binding energy (*Figure 3C*), in good agreement with previous studies. This Type 3 interaction was visualized in a structure of the SPARTIN MIT-IST1 complex (*Guo and Xu, 2015*). Ours is the first report of ESCRT-III binding by SNX15, and we found that both IST1 MIM elements contribute significantly, indicating that both elements can simultaneously occupy both the H1/H3 and H2/H3 grooves. SNX15 has been reported to function in endosomal protein trafficking (*Phillips et al., 2001*), which is also a major site of ESCRT function (see above).

As discussed in the main text, the MIT domains ULK1, VPS9D1 and NRBF2 did not bind ESCRT-III proteins in our screen.

