## [Editor Report]

The importance of the information provided to the community, their quality and the new questions they open will certainly make this paper an essential step forward in understanding the regulation of ESCRT-III.

---

## [Decision Letter]

**Decision letter after peer review:**

Thank you for submitting your article "Comprehensive analysis of the human ESCRT-III-MIT domain interactome reveals new cofactors for cytokinetic abscission" for consideration by *eLife*. Your article has been reviewed by 3 peer reviewers, one of whom is a member of our Board of Reviewing Editors, and the evaluation has been overseen by Vivek Malhotra as the Senior Editor. The following individuals involved in the review of your submission have agreed to reveal their identity: Jeremy G Carlton (Reviewer #2); Scott D Emr (Reviewer #3).

Overall, all reviewers agree on the importance of the findings reported here, not only for the field of ESCRT-III in cytokinesis, but for all the understanding of cellular functions in which ESCRT-III has an important role. The quality of the data, the exhaustivity of the in vitro study, and the new findings which confirm an important role of IST1 in all ESCRT-III functions are acknowledged by all reviewers. The reviewers support publication after a limited revision of the manuscript, which should focus on two main angles:

Essential revisions:

1) Expand the in vivo study on the role of enzymatic activities of SPASTIN, KATNA1 and CAPN7 in cytokinesis, following comments of Rev1 and Rev2:

Rev1:

"1-It is striking that SPASTIN and KATNA1 (Katanin), two MT-severing enzymes are required for completion of abscission, and localized to the midbody by the ESCRT-III complex. Of course, as shown by the Gerlich lab, they may be required to severe microtubules to allow for ESCRT-III constriction and membrane abscission. But SPASTIN is structurally very similar to VPS4, and was shown to also play a role in ESCRT-III function, in absence of MTs. I think a remaining question would be to know if the function of both SPASTIN and KATNA1 during cytokinesis is only linked to their enzymatic action on MTs, or whether some role in ESCRT-III turnover of both Spastin and/or Katanin could be responsible for their function in abscission. Maybe treating cells with taxol to stabilize MTs, and see how it modifies the recruitment of SPASTIN and KATANIN at the midbody could help. Also, it would be interesting to see if all ESCRT-III subunits are still recruited on midbody blocked by taxol, and see if the single mutants that disrupt specifically SPASTIN or KATANIN recruitment affect the recruitment of other subunits."

Rev2:

"A criticism of the manuscript is that the enzymatic activities delivered by these MIT-domain containing proteins to the midbody isn't explored. The manuscript could be strengthened by exploring whether the catalytic activity of CALPAIN7 was necessary for cytokinesis. Proteolytic control of ESCRT-III remodelling has been described recently (Tarrason-Risa et al., Science 2020) and whilst it is never nice to suggest new experimental directions in review, this might elevate the biological significance of the manuscript. "

2) Discuss technical aspects linked to fluorophore tagging in vitro and in vivo, as proposed by REv2 and Rev3:

Rev2:

"mCh-SPASTIN does not match the localisation described for endogenous SPASTIN (6A vs 5A) and only a limited pool of the mCh-CAPN7 localises to the midbody, compared to the endogenous stain. I appreciate that the authors highlight antibody accessibility as an issue, but an alternate explanation is that the mCh-tagged protein doesn't localise properly (this is an issue for many mCh-tagged proteins). In this light, the observation that only half of the midbodies displayed mCh-SPASTIN localisation (P10,L14) is a bit odd, as I would assume all midbodies would need to be SPASTIN positive; is GFP-SPASTIN localisation any clearer, does the position of the tag matter and if the authors divide this into early/late stage midbodies and arms/flemming body localisation, can any temporal changes in localiastion be quantified?"

Rev3:

"An explanation should be presented on why the authors believe the fluorophore-conjugation to ESCRT-III peptides does not have any effect on the affinity of interaction. The overall results of the paper, due to any changes in affinity because of the fluorophore, probably will not change, however if the authors are making strong quantitative arguments (as in Figure 2) to compare affinities of different interactions, some acknowledgement/experiments should be provided for this possibility. If experiments are pursued to disprove this possibility, a competition anisotropy assay (with unlabeled peptides), or some other technique like ITC that does not require labeling would be ideal. A few repeat measurements (perhaps of the strong binders that the authors characterize further) would be appropriate. Otherwise, there should at least be some mention of this limitation in the manuscript."

3) Discuss and expand the SPASTIN-IST1 vs SPASTIN-CHMP1B interactions

Rev1:

"2-One aspect that seems different from previous studies is the specific recruitment of SPASTIN, which seems to be more dependent on IST1 in the author's experiments, while it was shown to be specifically recruited by CHMP1b (the partner of IST1 in the ESCRT-III sequence) at the very end of abscission, in order to severe microtubules prior to membrane constriction. I wonder if the difference could be linked to differences in phosphorylation of the MIM domains of CHMP1B and IST1. Is there any way by which the authors could look at the phosphorylation of ESCRT-III subunits at the midbody? Could the author use their knowledge of specific MIT interactors to pull down specific subunits of the ESCRT-III complex and look at their phosphorylation when cells are chemically blocked during cytokinesis?"

Rev2:

"I was struck that the SPASTIN MIT domain binds the IST1 MIM 30x more strongly than the CHMP1B MIM (Figure 2), yet the interaction surface of the IST1 MIM (Figure 4A) is much greater than the CHMP1B MIM (Figure 4D). I think that these are both Type-1 MIMs; is there anything about the 'quality' of the contacts here that explains the differential affinities? Although the binding sites appear to overlap slightly, is there any competition for SPASTIN observable between CHMP1B and IST1 in-cells that is congruent with the affinity data?"

Please, also consider all relevant points made by the reviewers in their detailed review of your manuscript that you will find below.

*Reviewer #1 (Recommendations for the authors):*

1-It is striking that SPASTIN and KATNA1 (Katanin), two MT-severing enzymes are required for completion of abscission, and localized to the midbody by the ESCRT-III complex. Of course, as shown by the Gerlich lab, they may be required to severe microtubules to allow for ESCRT-III constriction and membrane abscission. But SPASTIN is structurally very similar to VPS4, and was shown to also play a role in ESCRT-III function, in absence of MTs. I think a remaining question would be to know if the function of both SPASTIN and KATNA1 during cytokinesis is only linked to their enzymatic action on MTs, or whether some role in ESCRT-III turnover of both Spastin and/or Katanin could be responsible of their function in abscission. Maybe treating cells with taxol to stabilize MTs, and see how it modifies the recruitment of SPASTIN and KATANIN at the midbody could help. Also, it would be interesting to see if all ESCRT-III subunits are still recruited on midbody blocked by taxol, and see if the single mutants that disrupt specifically SPASTIN or KATANIN recruitment affect the recruitment of other subunits.

2-One aspect that seems different from previous studies is the specific recruitment of SPASTIN, which seems to be more dependent on IST1 in the author's experiments, while it was shown to be specifically recruited by CHMP1b (the partner of IST1 in the ESCRT-III sequence) at the very end of abscission, in order to severe microtubules prior to membrane constriction. I wonder if the difference could be linked to differences in phosphorylation of the MIM domains of CHMP1B and IST1. Is there any way by which the authors could look at the phosphorylation of ESCRT-III subunits at the midbody? Could the author use their knowledge of specific MIT interactors to pull down specific subunits of the ESCRT-III complex and look at their phosphorylation when cells are chemically blocked during cytokinesis?

*Reviewer #2 (Recommendations for the authors):*

I find the biophysical and structural aspects of this manuscript strong and have little criticism here. One point that may be worth expanding is whether the interactions observed in a reductionist peptide/domain interaction hold true when considering the whole protein. I was struck that the SPASTIN MIT domain binds the IST1 MIM 30x more strongly than the CHMP1B MIM (Figure 2), yet the interaction surface of the IST1 MIM (Figure 4A) is much greater than the CHMP1B MIM (Figure 4D). I think that these are both Type-1 MIMs; is there anything about the 'quality' of the contacts here that explains the differential affinities? Although the binding sites appear to overlap slightly, is there any competition for SPASTIN observable between CHMP1B and IST1 in-cells that is congruent with the affinity data?

The microscopical data documenting localisation of endogenous KATNA1 & CAPN7 to the midbody was clear. I was less convinced by the mCherry-tagged transgene localisation; mCh-SPASTIN does not match the localisation described for endogenous SPASTIN (6A vs 5A) and only a limited pool of the mCh-CAPN7 localises to the midbody, compared to the endogenous stain. I appreciate that the authors highlight antibody accessibility as an issue, but an alternate explanation is that the mCh-tagged protein doesn't localise properly (this is an issue for many mCh-tagged proteins). In this light, the observation that only half of the midbodies displayed mCh-SPASTIN localisation (P10,L14) is a bit odd, as I would assume all midbodies would need to be SPASTIN positive; is GFP-SPASTIN localisation any clearer, does the position of the tag matter and if the authors divide this into early/late stage midbodies and arms/flemming body localisation, can any temporal changes in localiastion be quantified?

A criticism of the manuscript is that the enzymatic activities delivered by these MIT-domain containing proteins to the midbody isn't explored. The manuscript could be strengthened by exploring whether the catalytic activity of CALPAIN7 was necessary for cytokinesis. Proteolytic control of ESCRT-III remodelling has been described recently (Tarrason-Risa et al., Science 2020) and whilst it is never nice to suggest new experimental directions in review, this might elevate the biological significance of the manuscript.

The localisation of KATANIN to the midbody was interesting, but I think quantifying midbody arm enrichment relative to cytosol may be beneficial – the localisation here doesn't really look like previously described ESCRT localisations (in this or other manuscripts) and it would be good to exclude that the localisation isn't just cytosol in the midbody. Assuming it is genuine, there is an interesting localisation that isn't described in the manuscript; in Figure 5E, KATANIN can be seen at the terminus of the microtubule bundle, distal from the Flemming body. The same is true in Figure 6C; I think Echard has recently described this as an 'entry point' (Andrade et al., BioRxiv, 2021). Besides abscission in the midbody, it is possible that the post-spindle microtubule bundles need to be detached from the interphase microtubule network and I wonder if there was a role (perhaps ESCRT-independent?) that could be described for KATANIN here.

The degree of knockdown and phenotypic penetrance is weak. The authors claim that CAPN7 or KATANIN1 suppression mimics depletion of IST1 (P9 L14/15), however, the degree of IST1-depletion reported here (Figure 5 S1A) is limited. It is possible to deplete IST1 much more convincingly, and the author's reported effect on cytokinesis failure is weaker than previously observed (e.g., Agromayor et al., 2009), so I think it is true that CAPN7 or KATANIN depletion mimics a slight depletion of IST1, which is a slightly different interpretation of their involvement to that presented here.

*Reviewer #3 (Recommendations for the authors):*

1. An explanation should be presented on why the authors believe the fluorophore-conjugation to ESCRT-III peptides does not have any effect on the affinity of interaction. The overall results of the paper, due to any changes in affinity because of the fluorophore, probably will not change, however, if the authors are making strong quantitative arguments (as in Figure 2) to compare affinities of different interactions, some acknowledgement/experiments should be provided for this possibility. If experiments are pursued to disprove this possibility, a competition anisotropy assay (with unlabeled peptides), or some other technique like ITC that does not require labeling would be ideal. A few repeat measurements (perhaps of the strong binders that the authors characterize further) would be appropriate. Otherwise, there should at least be some mention of this limitation in the manuscript.

2. Somewhat related to #1, since many of the isotherms do not saturate, some of the numbers for the Kd measurements would be affected. Therefore, the authors should mention (especially in Figure 2) that many of the measurements are approximations of the Kd values.

3. We suggest the authors provide a model figure to help non-experts appreciate the complex set of interactions og ESCRT-III with MIT proteins, and how the recruitment of AAA+ ATPases to ESCRT-III polymers contributes to cytokinesis (and also in general to all ESCRT-related events).

---

## [Author Response]

Essential revisions:1) Expand the in vivo study on the role of enzymatic activities of SPASTIN, KATNA1 and CAPN7 in cytokinesis, following comments of Rev1 and Rev2:Rev1:"1-It is striking that SPASTIN and KATNA1 (Katanin), two MT-severing enzymes are required for completion of abscission, and localized to the midbody by the ESCRT-III complex. Of course, as shown by the Gerlich lab, they may be required to severe microtubules to allow for ESCRT-III constriction and membrane abscission. But SPASTIN is structurally very similar to VPS4, and was shown to also play a role in ESCRT-III function, in absence of MTs. I think a remaining question would be to know if the function of both SPASTIN and KATNA1 during cytokinesis is only linked to their enzymatic action on MTs, or whether some role in ESCRT-III turnover of both Spastin and/or Katanin could be responsible for their function in abscission. Maybe treating cells with taxol to stabilize MTs, and see how it modifies the recruitment of SPASTIN and KATANIN at the midbody could help. Also, it would be interesting to see if all ESCRT-III subunits are still recruited on midbody blocked by taxol, and see if the single mutants that disrupt specifically SPASTIN or KATANIN recruitment affect the recruitment of other subunits."

Reviewer 1 raises the intriguing hypothesis that in addition to severing microtubules, SPASTIN and KATNA1 may also contribute to abscission by remodeling ESCRT-III subunits. We are actually actively pursuing experiments aimed at testing this model, but reconstitution of ESCRT-mediated membrane fission and microtubule depolymerization in vitro are technically challenging experiments and we feel that the full range of experiments required to test this hypothesis rigorously is best performed as a separate study. We also note that Hurley and colleagues have recently reported that SPASTIN does not disassemble CHMP1B-IST1 coats from membrane nanotubes (Cada et al., 2022, bioRxiv). Whether both SPASTIN and KATNA1 need to sever microtubules, how their required activities differ (e.g., spatially vs. different types of microtubules), and whether one or both enzymes also depolymerize ESCRT-III filaments are all important questions, but we feel addressing these complex questions falls outside the scope of the current manuscript.

Rev2:"A criticism of the manuscript is that the enzymatic activities delivered by these MIT-domain containing proteins to the midbody isn't explored. The manuscript could be strengthened by exploring whether the catalytic activity of CALPAIN7 was necessary for cytokinesis. Proteolytic control of ESCRT-III remodelling has been described recently (Tarrason-Risa et al., Science 2020) and whilst it is never nice to suggest new experimental directions in review, this might elevate the biological significance of the manuscript."

We agree with this general thrust, and we are currently preparing a follow-up manuscript that more fully explores the enzymatic activity of CAPN7.

2) Discuss technical aspects linked to fluorophore tagging in vitro and in vivo, as proposed by REv2 and Rev3:Rev2:"mCh-SPASTIN does not match the localisation described for endogenous SPASTIN (6A vs 5A) and only a limited pool of the mCh-CAPN7 localises to the midbody, compared to the endogenous stain. I appreciate that the authors highlight antibody accessibility as an issue, but an alternate explanation is that the mCh-tagged protein doesn't localise properly (this is an issue for many mCh-tagged proteins). In this light, the observation that only half of the midbodies displayed mCh-SPASTIN localisation (P10,L14) is a bit odd, as I would assume all midbodies would need to be SPASTIN positive; is GFP-SPASTIN localisation any clearer, does the position of the tag matter and if the authors divide this into early/late stage midbodies and arms/flemming body localisation, can any temporal changes in localiastion be quantified?"

Reviewer 2 raises the valid concern that tagging SPASTIN with mCherry could affect its localization. To address this concern, we made two new SPASTIN constructs, each tagged with a single short HA epitope at either the N- or C-terminus, and we created stable cell lines that expressed these new constructs, as well as the mCh-SPASTIN constructs. Using these lines, we compared HA-SPASTIN localization (detected by anti-HA antibodies) with localization of mCh-SPASTIN (detected by anti-mCherry antibodies) in both the original cell line used in the manuscript (mCh-SPASTIN #22) and a new cell line that was independently generated using the same mCh-SPASTIN construct. Quantification of localization from three independent experiments in all four cell lines appeared very similar (see Response Figure 2, below), demonstrating that mCherry-tagged SPASTIN localization is similar to that of SPASTIN constructs with short epitope tags at either terminus.

We recognize that there are some differences between the detailed distribution of midbody localization (Flemming body vs. Flemming Body + Arms) in the comparison of endogenous and exogenous SPASTIN (and we reported these differences for transparency). The differences may reflect differences in expression level or other factors. Differences in assay sensitivity can change the total number of detectable midbodies, and in the more recent experiments shown in Author response image 1 , the percent of detectable midbodies was higher than Manuscript Figure 6, even when probing mCherry as before. To overcome these limitations, we confined our analysis in Manuscript Figure 6 to comparisons of the total midbody recruitment of exogenously expressed proteins, and we used relative comparisons to determine the effect of point mutations on midbody localization. The assay worked well for this purpose, the differences are dramatic, so we believe that the data conclusively demonstrate that the point mutations that disrupt ESCRT-III binding disrupt midbody localization (Manuscript Figure 6).

Further information on CAPN7 is provided above, where the most important point is that exogenously expressed CAPN7 rescues the effects of CAPN7 depletion on abscission and the NoCut response, and again the localization assay robustly reports on requirements for targeting CAPN7 to the midbody.

Rev3:"An explanation should be presented on why the authors believe the fluorophore-conjugation to ESCRT-III peptides does not have any effect on the affinity of interaction. The overall results of the paper, due to any changes in affinity because of the fluorophore, probably will not change, however if the authors are making strong quantitative arguments (as in Figure 2) to compare affinities of different interactions, some acknowledgement/experiments should be provided for this possibility. If experiments are pursued to disprove this possibility, a competition anisotropy assay (with unlabeled peptides), or some other technique like ITC that does not require labeling would be ideal. A few repeat measurements (perhaps of the strong binders that the authors characterize further) would be appropriate. Otherwise, there should at least be some mention of this limitation in the manuscript."

Reviewer #3 raises the valid point that fluorophores have the potential to alter binding between ESCRT-III proteins and purified MIT proteins.

1) As the reviewer recommended, we have added text mentioning this limitation on page 5, line 21. We now say:

“Our screen generally recapitulated binding interactions reported previously using orthogonal techniques (See Supplemental Discussion), but we cannot rule out the possibility that in some cases the position of the label could have interfered with or artificially enhanced binding interactions for some ESCRT-III-MIT pairs.”

2) Below, we provide several additional lines of experimental evidence indicating that the fluorophores did not grossly influence our binding results:

a. As suggested by Reviewer #3, we measured and report binding of labeled and unlabeled CHMP4B with MITD1 and USP8, and IST1 with UPS8. As shown in Author response table 1 (and provided as Figure 2—figure supplement 5B-E), there is generally reasonable agreement between the different K_D_ values measured by direct binding (fluorophore present) vs. competition (fluorophore absent).

**Author response table 1. sa2table1:** 

MIT Domain	ESCRT-III Tail	K_D_ (direct binding) (µM)	K_I_ competition experiment (µM)
MITD1	CHMP4B	15 ± 1 (n≥3)	21.7 ± 0.4 (n≥3)
USP8	CHMP4B	11.7 ± 0.9 (n≥3)	38.9 ± 0.3 (n=2; error is the range of the measurement)
USP8	IST1-MIM1/3	0.6 ± 0.01 (n≥3)	1.1 ±- 0.2 (n=2; error is the range of the measurement)

b. In the case of IST1 binding to CAPN7, we compared binding in which the fluor was on the IST1 MIM N- and C-termini and found that having the fluor at different positions did not affect the measured binding affinities:

**Author response table 2. sa2table2:** 

MIT Domain	ESCRT-III Tail	K_D_ (N-terminal Fluor) (µM)	K_D_ (C-terminal Fluor) (µM)
CAPN7	IST1	0.07 ± 0.02 (n≥3)	0.09 ± 0.01 (n=3)

c. For the SPASTIN MIT-IST1 interaction, we compared the results of fluorescence polarization anisotropy (FP) binding assays (FP detection, fluorophore present) to biosensor binding assays (SPR detection, no fluorophore). K_D_ values determined using these orthogonal approaches were again very similar.

**Author response table 3. sa2table3:** 

MIT Domain	ESCRT-III Tail	K_D_ (FP) (µM)	K_D_ (SPR) (µM)
SPASTIN	IST1	0.5 ± 0.1 (n≥ 3)	0.57 ± 0.08 (n=2)

3) Discuss and expand the SPASTIN-IST1 vs SPASTIN-CHMP1B interactionsRev1:"2-One aspect that seems different from previous studies is the specific recruitment of SPASTIN, which seems to be more dependent on IST1 in the author's experiments, while it was shown to be specifically recruited by CHMP1b (the partner of IST1 in the ESCRT-III sequence) at the very end of abscission, in order to severe microtubules prior to membrane constriction. I wonder if the difference could be linked to differences in phosphorylation of the MIM domains of CHMP1B and IST1. Is there any way by which the authors could look at the phosphorylation of ESCRT-III subunits at the midbody? Could the author use their knowledge of specific MIT interactors to pull down specific subunits of the ESCRT-III complex and look at their phosphorylation when cells are chemically blocked during cytokinesis?"

An important point is that our observation that CHMP1B binding is not necessary for SPASTIN recruitment is not, in fact, in conflict with the actual data reported by Hurley and colleagues (though the conclusions that we derive from the data do differ).

The original assertion that CHMP1B was the primary recruiter of SPASTIN to the midbody was based on the observation that (1) CHMP1B depletion by siRNA blocked SPASTIN recruitment, and (2) the SPASTIN mutation F124D, H120D, blocked localization to the midbody (Yang et al., 2008).

In fact, the IST1 requirement for SPASTIN midbody localization that we observe was not ruled out by these studies (they did not examine IST1 at all), particularly as subsequent experiments have shown that:

1) Loss of CHMP1 (and other ESCRT-III) alters IST1 localization, producing aberrant midbody-localized structures (Goliand *et al.*, 2017, *Cell Reports*), and blocks recruitment to endosomes entirely (Dimaano *et al.* 2008, Rue *et al.*, 2008). Hence, the Yang et al. experiments that depleted CHMP1B would also have altered IST1 recruitment, and therefore did not cleanly report solely on the requirement for CHMP1B.

2) Our IST1 binding data (and structure) show that the F124D mutation used by Yang et al. blocks the binding of IST1, CHMP1B and CHMP3 (Manuscript; Figure 4 and Figure 4—figure supplement 2). Hence, they were not specifically testing the role of CHMP1B in SPASTIN recruitment (and we agree that this mutation does block SPASTIN recruitment, we just attribute that block to loss of IST1 binding, not loss of CHMP1B binding).

By virtue of our selective SPASTIN-CHMP1B mutant L177D, we can now say for the first time, that *neither* CHMP1B nor CHMP3 binding are required for SPASTIN localization when IST1 is present.

Although we agree that phosphorylation (or any PTM) of ESCRT-III proteins during cytokinesis is an interesting potential mechanism for regulating ESCRT-III-MIT partnerships, we don’t believe that it is necessary to invoke this model to explain our data and we feel these experiments are beyond the scope of the paper.

Rev2:"I was struck that the SPASTIN MIT domain binds the IST1 MIM 30x more strongly than the CHMP1B MIM (Figure 2), yet the interaction surface of the IST1 MIM (Figure 4A) is much greater than the CHMP1B MIM (Figure 4D). I think that these are both Type-1 MIMs; is there anything about the 'quality' of the contacts here that explains the differential affinities? Although the binding sites appear to overlap slightly, is there any competition for SPASTIN observable between CHMP1B and IST1 in-cells that is congruent with the affinity data?"

The reviewer raises a good point – this really is a case where buried surface area does not correlate well with binding affinity (which is actually quite common). We believe that the tighter IST1 binding instead likely arises from differences in the composition of hydrophobic residues that are buried in the H1/H3 groove of the SPASTIN MIT domain. IST1 buries 2 aromatics and 2 aliphatics, whereas CHMP1B buries only 3 aliphatics. Additionally, the additional ordering of the longer CHMP1B helix will reduce entropy and also contribute to the loss of binding energy. To address this issue, we have added to the manuscript the following text that makes these points (Page 8 Line 16):

“IST1 binding is slightly tighter than CHMP1B binding, likely owing to enhanced hydrophobic interactions with the SPASTIN H1/H3 groove (Figure 4—figure supplement 1C). IST1 buries two aromatic and two aliphatic side chains, whereas the CHMP1B binding element lacks aromatic residues and buries only three aliphatics.”

Our structure reveals that the IST1 binding site on SPASTIN overlaps completely with the CHMP1B binding site (but not vice versa – see Manuscript; Figure 4—figure supplement 1). Hence, both ESCRT-III proteins cannot bind individual SPASTIN subunits simultaneously. However, SPASTIN functions as a hexamer, so we cannot rule out the possibility that hexamers can bind both IST1 and CHMP1B simultaneously. Although we don’t have a way to measure competition between ESCRT-III for SPASTIN binding in cells, we believe that the identification and deployment of the L177D SPASTIN mutation to selectively disrupt the CHMP1B and CHMP3 interactions in cells represents a significant advance in our understanding of the functional requirements for specific MIT-ESCRT-III partnerships.

A final important point is that we noticed that there are known human mutations near the SPASTIN MIT domain that cause spastic paraplegia, but they lie just outside the construct that we were using in our binding studies (which was based on the crystal structure from the Hurley lab). We therefore tested the binding of a construct that was extended slightly on both ends of our previous minimal construct. Indeed, we found that the slightly longer SPASTIN MIT construct (residues 108-200) bound ~5-fold more tightly to CHMP1B (K_D_ = 2.3 µM; Figure 2) compared to the minimal MIT sequence (residues 112-196; CHMP1B K_D_ ~ 12-15 µM; data not shown and Yang et al., 2008). Moreover, we now observe weak but significant CHMP3 binding (K_D_ = 59 µM) (Figure 2, Figure 3A) and mapped this binding to the H1-H3 groove of SPASTIN (Figure 4—figure supplement 2; new figure added in revision). The upshot of all of this is that we are now reporting binding data for the longer SPASTIN MIT domain (Figure 2, Figure 3A, and Figure 4E and F, Figure 4—figure supplement 2). The affinity difference between IST1 and CHMP1B binding is only 5-fold, and we now report CHMP3 as a SPASTIN binding partner.

Reviewer #2 (Recommendations for the authors):The localisation of KATANIN to the midbody was interesting, but I think quantifying midbody arm enrichment relative to cytosol may be beneficial – the localisation here doesn't really look like previously described ESCRT localisations (in this or other manuscripts) and it would be good to exclude that the localisation isn't just cytosol in the midbody. Assuming it is genuine, there is an interesting localisation that isn't described in the manuscript; in Figure 5E, KATANIN can be seen at the terminus of the microtubule bundle, distal from the Flemming body. The same is true in Figure 6C; I think Echard has recently described this as an 'entry point' (Andrade et al., BioRxiv, 2021). Besides abscission in the midbody, it is possible that the post-spindle microtubule bundles need to be detached from the interphase microtubule network and I wonder if there was a role (perhaps ESCRT-independent?) that could be described for KATANIN here.

To confirm the intriguing localization found for KATNA1 shown in the manuscript, we have now compared two independent KATNA1 antibodies (Proteintech – 17560-1-AP; and Abcam – ab111881) and additionally compared two different fixation conditions: PFT (1xPHEM + 4%PFA + 0.5%Triton) vs. Methanol. These data are provided in Author response image 1. Importantly, the patterns were all very similar with both antibodies and under both fixation conditions. We have now added this point to the experimental methods (Page 32, Line 14):

“In the case of KATNA1, localization was confirmed with two independent antibodies (Proteintech – #17560-1-AP; Abcam – ab111881) and two different fixation conditions: -20°C Methanol for 10 min and PFT (1xPHEM+4%PFA+0.5%Triton as described above) (data not shown).”

**Author response image 1. sa2fig1:** Cells were treated as in Manuscript Figure 5D-H (active checkpoint), fixed in either PFT or Methanol and stained with KATNA1-specific antibodies from Proteintech or Abcam. Two examples from each treatment are shown.

We also thank the reviewer for pointing out the resemblance to the recently described “midbody entry point” and we have added text to the manuscript describing this localization with a reference to the Echard paper. Page 9, line 5:

“Uniquely, KATNA1 also distributed along the midbody arms, reminiscent of localization observed for caveolae, and termed “midbody entry points” (Andrade et al., 2022).” Author response image 1

The degree of knockdown and phenotypic penetrance is weak. The authors claim that CAPN7 or KATANIN1 suppression mimics depletion of IST1 (P9 L14/15), however, the degree of IST1-depletion reported here (Figure 5 S1A) is limited. It is possible to deplete IST1 much more convincingly, and the author's reported effect on cytokinesis failure is weaker than previously observed (e.g., Agromayor et al., 2009), so I think it is true that CAPN7 or KATANIN depletion mimics a slight depletion of IST1, which is a slightly different interpretation of their involvement to that presented here.

The reviewer is correct. In general, our MIT protein knockdown phenotypes approximate a moderate knockdown of IST1. We’ve added the following text on page 9, line 38:

“These pronounced phenotypes resembled the abscission defects observed with moderate knockdown of the essential IST1 protein (positive control) (Bajorek et al., 2009a, Agromayor et al., 2009)

Reviewer #3 (Recommendations for the authors):1. Somewhat related to #1, since many of the isotherms do not saturate, some of the numbers for the Kd measurements would be affected. Therefore, the authors should mention (especially in Figure 2) that many of the measurements are approximations of the Kd values.

We agree, and we have added an explanation that are K_D_s are approximations, and a description of how we generated our cutoffs/analyses as follows: Page 5, line 25:

“Pairwise binding isotherms for weak binding pairs often did not reach saturation (Figure 2—figure supplements 1-3), and the K_D_s for these interactions should therefore be considered approximations. In cases where isotherms did not reach half saturation at the highest MIT concentration tested (~100 µM), we did not attempt to estimate the K_D_.”

2. We suggest the authors provide a model figure to help non-experts appreciate the complex set of interactions og ESCRT-III with MIT proteins, and how the recruitment of AAA+ ATPases to ESCRT-III polymers contributes to cytokinesis (and also in general to all ESCRT-related events).

We agree that the multitude and combinatorial nature of MIT-ESCRT-III interactions is complex. However, given that the depth of our understanding of these interactions varies from atomic resolution (SPASTIN) to initial discovery (KATNA1), we do not yet feel that we have reached the point where we can make a model figure describing ATPase function that would aid in the interpretation of our data. We are hopeful that future studies, like those in progress for CAPN7, will add mechanistic details to the groundwork laid here by our initial identification of the MIT-ESCRT-III network, and we will feel more comfortable providing model figures at that point.